# Adaptive learning and decision-making under uncertainty by metaplastic synapses guided by a surprise detection system

Kiyohito Iigaya[1,2,3]*

[1]Gatsby Computational Neuroscience Unit, University College London, London, United Kingdom; [2]Center for Theoretical Neuroscience, College of Physicians and Surgeons, Columbia University , New York, United States; [3]Department of Physics, Columbia University, New York, United States

**Abstract** Recent experiments have shown that animals and humans have a remarkable ability to adapt their learning rate according to the volatility of the environment. Yet the neural mechanism responsible for such adaptive learning has remained unclear. To fill this gap, we investigated a biophysically inspired, metaplastic synaptic model within the context of a well-studied decision-making network, in which synapses can change their rate of plasticity in addition to their efficacy according to a reward-based learning rule. We found that our model, which assumes that synaptic plasticity is guided by a novel surprise detection system, captures a wide range of key experimental findings and performs as well as a Bayes optimal model, with remarkably little parameter tuning. Our results further demonstrate the computational power of synaptic plasticity, and provide insights into the circuit-level computation which underlies adaptive decision-making.

*For correspondence: kiigaya@ gatsby.ucl.ac.uk

**Competing interests:** The author declares that no competing interests exist.

## Introduction

From neurons to behavior, evidence shows that adaptation takes place over a wide range of timescales, with temporal dynamics often captured by power-law, or collection of multiple exponents, rather than a single exponent (*Thorson and Biederman-Thorson, 1974*; *Ulanovsky et al., 2004*; *Corrado et al., 2005*; *Fusi et al., 2007*; *Kording et al., 2007*; *Wark et al., 2009*; *Lundstrom et al., 2010*; *Rauch et al., 2003*; *Pozzorini et al., 2013*). On the other hand, single-exponent model analysis of behavioral data showed that the time constant of exponents (or learning rate) changed across trials (*Behrens et al., 2007*; *Rushworth and Behrens, 2008*; *Soltani et al., 2006*; *Nassar et al., 2010*; *Nassar et al., 2012*; *Neiman and Loewenstein, 2013*; *McGuire et al., 2014*). While theoretical and experimental studies strongly suggest that activity-dependent synaptic plasticity plays a crucial role in learning and adaptation in general (*Martin et al., 2000*; *Kandel et al., 2000*; *Dayan and Abbott, 2001*), the neural mechanisms behind flexible learning, especially in the case of decision making under uncertainty, has remained unclear. To address this issue, here we investigate the roles of synaptic plasticity within an established decision-making neural circuit model, and propose a model that can account for empirical data.

Standard learning models which use a single learning rate, $\alpha$, fail to capture multiple timescales of adaptation, including those described by a power-law, since these models can only store and update memory on a single timescale of $\tau \sim 1/\alpha$. This includes a well studied switch-like synaptic model of memory (*Amit and Fusi, 1994*; *Fusi and Abbott, 2007*) in which synapses make transitions between weak- and strong-efficacy states at a rate $\alpha$. It has been shown that its transition rate $\alpha$

**eLife digest** Humans and other animals have a remarkable ability to adapt their decision making to changes in their environment. An experiment called the "multi-armed bandit task" shows this process in action. The individual's role in this task is to choose between multiple targets. One of these has a higher probability of reward than the other three, and individuals soon begin to favor this target over the others. If the identity of the most rewarded target changes, individuals adjust their responses accordingly. Crucially, however, individuals learn more quickly when the identity of the most rewarded target changes frequently. In other words, they learn faster in an uncertain world.

Changes in the strength of connections between neurons – called synapses – are thought to underlie such learning processes. Receiving a reward strengthens synapses in a process referred to as synaptic plasticity. However, the standard model of synaptic plasticity – in which synapses change from weak to strong or vice versa at a constant rate – struggles to explain why individuals learn more quickly under variable conditions.

An alternative model of learning is the cascade model, which incorporates 'metaplasticity'. This assumes that the rateof synaptic plasticity can also vary; that is, synapses change their strength at different speeds. The cascade model is based on the observation that multiple biochemical signaling cascades contribute to synaptic plasticity, and some of these are faster than others. Kiyohito Iigaya therefore decided to test whether the cascade model could explain data from experiments such as the four-armed bandit task. While the cascade model was indeed more flexible than the standard model of synaptic plasticity, it still could not fully explain the observed results.

Iigaya solved the problem by introducing an external "surprise detection system" into the model. Doing so enabled the model to detect a sudden change in the environment and to rapidly increase the rate of learning, just as individuals do in real life. The surprise detection system allowed synapses to quickly forget what they had learned before, which in turn made it easier for them to engage in new learning. The next step is to identify the circuit behind the surprise detection system: this will require further theoretical and experimental studies.

effectively functions as the learning rate of systems with populations of such synapses in a decision making network (*Soltani and Wang, 2006*; *Fusi et al., 2007*; *Iigaya and Fusi, 2013*). Unlike the classical un-bounded synapses model, this switch-like model incorporates a biologically relevant assumption of bounded synaptic weights. However, by itself, the plausible assumption of bounded synapses fails to capture key phenomena of adaptive learning, including well-documented multiple timescales of adaptation (*Thorson and Biederman-Thorson, 1974*; *Ulanovsky et al., 2004*; *Corrado et al., 2005*; *Fusi et al., 2007*; *Kording et al., 2007*; *Wark et al., 2009*; *Lundstrom et al., 2010*; *Rauch et al., 2003*; *Pozzorini et al., 2013*).

It is however known that there are various chemical cascade processes taking place in synapses that affect synaptic plasticity (*Citri and Malenka, 2008*; *Kotaleski and Blackwell, 2010*). Those processes, in general, operate on a wide range of timescales (*Zhang et al., 2012*; *Kramar et al., 2012*). To capture this complex, multi-timescale synaptic plasticity in a minimum form, a complex – but still switch-like – synaptic model, the *cascade model* of synapses, has been proposed (*Fusi et al., 2005*). In the cascade model, synapses are still bounded in their strengths but assumed to be *metaplastic*, meaning that, in addition to the usual case of adaptable synaptic strengths, synapses are also permitted to change their rates of plasticity $\alpha$. The resulting model can efficiently capture the widely-observed power-law forgetting curve (*Wixted and Ebbesen, 1991*). However, application has been limited to studies of the general memory storage problem (*Fusi et al., 2005*; *Savin et al., 2014*), where synapses *passively* undergo transitions in response to uncorrelated learning events.

Indeed, recent experiments show that humans and other animals have a remarkable ability to *actively* adapt themselves to changing environments. For instance, animals can react rapidly to abrupt step-like changes in environments (*Behrens et al., 2007*; *Rushworth and Behrens, 2008*; *Soltani et al., 2006*; *Nassar et al., 2010*; *Nassar et al., 2012*; *Neiman and Loewenstein, 2013*; *McGuire et al., 2014*), or change their strategies dynamically (*Summerfield et al., 2011*;

*Donoso et al., 2014*). While the original cascade model (*Fusi et al., 2005*) is likely to be able to naturally encode multiple timescales of reward information (*Corrado et al., 2005*; *Fusi et al., 2007*; *Bernacchia et al., 2011*; *Iigaya et al., 2013*; *Iigaya, 2013*), such *active* adaptation may also require external guidance, such as in the form of a surprise signal (*Hayden et al., 2011*; *Garvert et al., 2015*).

So far the computational studies of such changes in learning rates have largely been limited to optimal Bayesian inference models (e.g. *Behrens et al., 2007*). While those models can account for normative aspects of animal's inference and learning, they provide limited insight into how probabilistic inference can be implemented in neural circuits.

To address these issues, in this paper we apply the cascade model of synapses to a well studied decision-making network. Our primary finding is that the cascade model of synapses can indeed capture the remarkable flexibility shown by animals in changing environments, but under the condition that synaptic plasticity is guided by a novel surprise detection system with simple, non-cascade type synapses. In particular, we show that while the cascade model of synapses is able to consolidate reward information in a stable environment, it is severely limited in its ability to adapt to a sudden change in the environment. The addition of a surprise detection system, which is able to detect such abrupt changes, facilitates adaptation by enhancing the synaptic plasticity of the decision-making network. We also shows that our model can capture other aspects of learning, such as spontaneous recovery of preference (*Mazur, 1996*; *Gallistel et al., 2001*).

## Results

### The tradeoff in the rate of synaptic plasticity under uncertainty in decision making tasks

In this paper, we analyze our model in stochastically-rewarding choice tasks in two slightly different reward schedules. One is a concurrent variable interval (VI) schedule, where rewards are given stochastically according to fixed contingencies. Although the optimal behavior is to repeat a

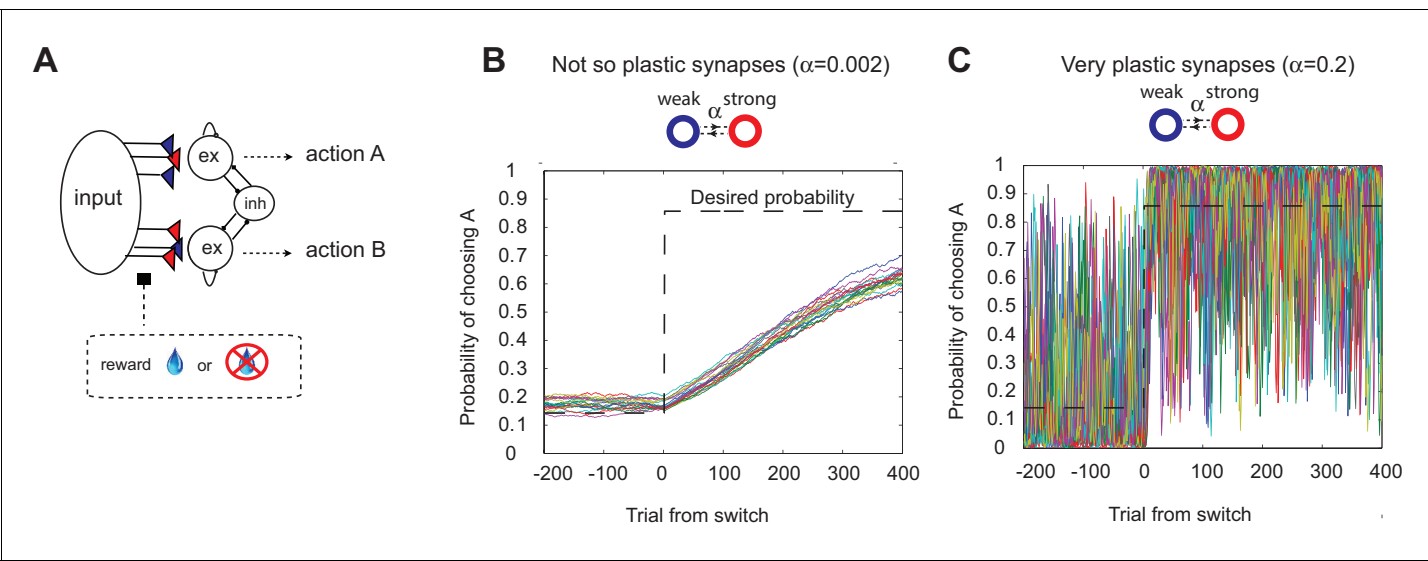

**Figure 1.** The decision making network and the speed accuracy tradeoff in synaptic learning. (**A**) The decision making network. Decisions are made based on the competition (winner take all process) between the excitatory action selective populations, via the inhibitory population. The winner is determined by the synaptic strength between the input population and the action selective populations. After each trial, the synaptic strength is modified according to the learning rule. (**B**, **C**). The speed accuracy tradeoff embedded in the rate of synaptic plasticity. The horizontal dotted lines are the ideal choice probability and the colored lines are different simulation results under the same condition. The vertical dotted lines show the change points, where the reward contingencies were reversed. The choice probability is reliable only if the rate of plasticity is set to be very small ($\alpha = 0.002$); however, then the system cannot adjust to a rapid unexpected change in the environment (**B**). On the other hand, highly plastic synapses ($\alpha = 0.2$) can react to a rapid change, but with a price to pay as a noisy estimate afterwards (**C**).

deterministic choice sequence according to the contingencies, animals instead show probabilistic choices described by the matching law (*Herrnstein, 1961*; *Sugrue et al., 2004*; *Lau and Glimcher, 2005*) in which the fraction of choices is proportional to the fraction of rewards obtained from the choice. In fact, the best probabilistic behavior under this schedule is to throw a dice with a bias given by the matching law (*Sakai and Fukai, 2008*; *Iigaya and Fusi, 2013*). We therefore assume that the goal of subjects in this case is to implement the matching law, which has previously been shown to be produced by the model under study (*Soltani and Wang, 2006*; *Fusi et al., 2007*; *Wang, 2008*; *Iigaya and Fusi, 2013*). The other schedule is a variable rate (VR) schedule, also known as a multi-armed bandit task, where the probability of obtaining a reward is fixed for each choice. In this case, subjects need to figure out which choice currently has the highest probability of rewards. In both tasks, subjects are required to make adaptive decision making according to the changing values of options in order to collect more rewards.

We study the role of synaptic plasticity in a well-studied decision making network (*Soltani and Wang, 2006*; *Fusi et al., 2007*; *Wang, 2008*; *Iigaya and Fusi, 2013*) illustrated in *Figure 1A*. The network has three types of neural populations: (1) an input population, which we assume to be uniformly active throughout each trial; (2) action selection populations, through which choices are made; and (3) an inhibitory population, through which different action selection populations compete. It has been shown that this network shows attractor dynamics with bi-stability, corresponding to a winner-take-all process acting between action selection populations. We assume that choice corresponds to the winning action selection population, as determined by the synaptic strength projecting from input to action selection populations. It has been shown that the decision probability can be well approximated by a sigmoid of the difference between the strength of two synaptic populations $E_A$ and $E_B$ (*Soltani and Wang, 2006*):

$$P_A = \frac{1}{e^{-\frac{E_A - E_B}{T}} + 1},$$

(1)

where $P_A$ is the probability of choosing target $A$, and the temperature $T$ is a free parameter describing the noise in the network.

This model can show adaptive probabilistic choice behaviors when assuming simple reward-based Hebbian learning (*Soltani and Wang, 2006*, *2010*; *Iigaya and Fusi, 2013*). We assume that the synaptic efficacy is bounded, since this has been shown to be an important biologically-relevant assumption (*Amit and Fusi, 1994*; *Fusi and Abbott, 2007*). As the simplest case, we assume binary synapses, and will call states 'depressed' and 'potentiated', with associated strengths 0 (weak) and 1 (strong), respectively. We previously showed that the addition of intermediate synaptic efficacy states does not alter the model's performance (*Iigaya and Fusi, 2013*). At the end of each trial, synapses are modified stochastically depending on the activity of the pre- and post-synaptic neurons and on the outcome (i.e. whether the subject receives a reward or not). The synapses projecting from the input population to the winning target population are potentiated stochastically with probability $\alpha_r$ in case of a reward, while they are depressed stochastically with probability $\alpha_{nr}$ in case of no-reward (for simplicity we assume $\alpha_r = \alpha_{nr} = \alpha$, otherwise explicitly noted). These transition probabilities are closely related to the plasticity of synapses, as a synapse with a larger transition probability is more vulnerable to changes in strength. Thus, we call $\alpha$'s the rate of plasticity. The total synaptic strength projecting to each action selection population encodes the reward probability over the timescale of $1/\alpha$ (*Soltani and Wang, 2006*; *Soltani and Wang, 2010*; *Iigaya and Fusi, 2013*) (For more detailed learning rules, see the Materials and methods section).

It has also been shown, however, that this model exhibits limited flexibility in the face of abrupt changes of timescales in the environment (*Soltani and Wang, 2006*; *Iigaya and Fusi, 2013*). This is due to the trade-off: a high rate of synaptic plasticity is necessary to react to a sudden change, but at the cost of very noisy estimation (as the synapses inevitably track local noise). This is illustrated in *Figure 1B,C*, where we simulated our model with a fixed rate of synaptic plasticity in a VI reward schedule in which reward contingencies change abruptly (*Sugrue et al., 2004*; *Corrado et al., 2005*). As seen in *Figure 1B,C*, the choice probability is reliable only if the rate of plasticity is set to be very small ($\alpha = 0.002$); however, then the system cannot adjust to a rapid unexpected change in the environment (*Figure 1B*). On the other hand, highly plastic synapses ($\alpha = 0.2$) can react to a rapid change, but with a price to pay as a noisy estimate afterwards (*Figure 1C*).

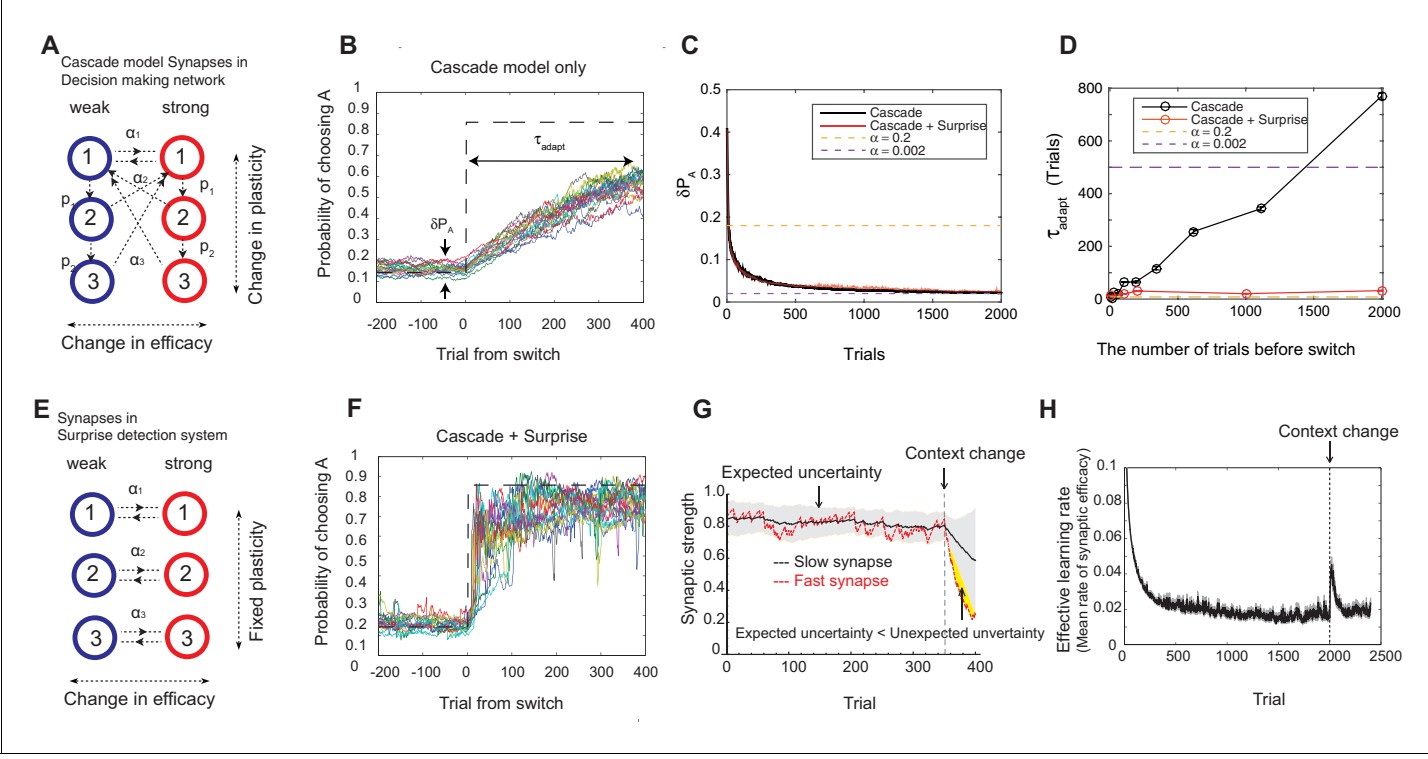

**Figure 2.** Our model solves the tradeoff the cascade model of metaplastic synapses guided by a surprise detection system. (A) The cascade model of synapses for the decision making network. The synaptic strength is assumed to be binary (weak or strong); and there are multiple (three for each strength, in this example) meta-plastic states associated with these strengths. The transition probability of changing synaptic strength is denoted by $\alpha_i$, while the transition probability of changing plasticity itself is denoted by $p_i$, where $\alpha_1 > \alpha_2 > ...$ and $p_1 > p_2 > ....$ Deeper states are less plastic and less likely to enter. (B) The cascade model of synapses can reduce the fluctuation of estimation when the environment is stationary, thanks to the memory consolidation; however, the model fails to respond to a sudden change in the environment. (C) The changes in the fluctuation of choice probability in a stable environment. The cascade model synapses (black) can reduce the fluctuation gradually over time. This is also true when a surprise detection network (described below) is present. The dotted lines indicate the case with a single fixed plasticity that are used in *Figure 1B,C*. The probability fluctuation $\delta P_A$ is defined as a mean standard deviation in the simulated choice probabilities. The synapses are assumed to be at the most plastic states at $t = 0$. (D) The adaptation time required to switch to a new environment after a change point as a function of the size of the previous stable environment. The adaptation time increases proportionally to the duration of the previous stable environment for the cascade model (black). The surprise detection network can significantly reduce the adaptation time independent of the previous context length (red). The adaptation time $\tau$ is defined as the number of trials required to cross the threshold probability ($P_A = 0.7$) after the change point. (E) The simple synapses in the surprise detection network. Unlike the cascade model, the rate of plasticity is fixed, and each group of synapses takes one of the logarithmically segregated rates of plasticity $\alpha_i$'s. (F) The decision making network with the surprise detecting system can adapt to an unexpected change. (G) How a surprise is detected. Synapses with different rates of plasticity encode reward rates on different timescales (only two are shown). The mean difference between the reward rates (expected uncertainty) is compared to the current difference (unexpected uncertainty). A surprise signal is sent when the unexpected uncertainty significantly exceeds the expected uncertainty. The vertical dotted line shows the change point, where the reward contingency is reversed. (H) Changes in the mean rates of plasticity (effective learning rate) in the cascade model with a surprise signal. Before the change point in the environment, the synapses become gradually less and less plastic; but after the change point, thanks to the surprise signal, the cascade model synapses become more plastic. In this figure, the network parameters are taken as $\alpha_i = (\frac{1}{5})^i$, $p_i = (\frac{1}{5})^i$, $T = 0.1$, $\gamma = 0$, $m = 10$, $h = 0.05$, while the total baiting probability is set to 0.4 and the baiting contingency is set to 9 : 1 (VI schedule).

## Changing plasticity according to the environment: the cascade model of synapses and the surprise detection system

How can animals solve this tradeoff? Experimental studies suggest that they integrate reward history on multiple timescales rather than a single timescale (*Corrado et al., 2005*; *Fusi et al., 2007*; *Bernacchia et al., 2011*). Other studies show that animals can change the integration timescale, or the learning rate, depending on the environment (*Behrens et al., 2007*; *Nassar et al., 2010*; *Nassar et al., 2012*). To incorporate these findings into our model, we use a synaptic model that can change the rate of plasticity $\alpha$ itself, in addition to the strength (weak or strong), depending on

the environment. The best known and successful model is the cascade model of synapses, originally proposed to incorporate biochemical cascade process taking place over a wide range of timescales (*Fusi et al., 2005*). In the cascade model, illustrated in *Figure 2A*, the degree of synaptic strength is still assumed to be binary (weak or strong); however, there are $m$ states with different levels of plasticity $\alpha_1, \alpha_2, \ldots, \alpha_m$, where $\alpha_1 > \alpha_2 > \ldots > \alpha_m$. The model also allows transitions from one level of plasticity to another with a *metaplastic* transition probability $p_i$ ($i = 1, 2, \ldots, m-1$) that is fixed depending on the depth. Following (*Fusi et al., 2005*), we assume $p_1 > p_2 > \ldots > p_{m-1}$, meaning that entering less plastic states becomes less likely to occur with increasing depth. All the transitions follow the same reward-based learning rule with corresponding probabilities, where the probabilities are separated logarithmically (ex. $\alpha_i = \left(\frac{1}{2}\right)^i$ and $p_i = \left(\frac{1}{2}\right)^i$ ) following (*Fusi et al., 2005*) (see Materials and methods section for more details).

We found that the cascade model of synapses can encode reward history on a wide, variable range of timescales. The wide range of transition probabilities in the model allows the system to encode values on multiple timescales, while the meta-plastic transitions allow the model to vary the range of timescales. These features allow the model to consolidate the value information in a steady environment, as the synapses can become less plastic (*Figure 2B–D*). As seen in *Figure 2C*, the fluctuation of choice probability with the cascade model synapses becomes smaller as the model stays in the stable environment, where we artificially set that all synapses are initially at the most plastic states (top states). Because of the reward-based metaplastic transitions, more and more synapses gradually occupy less plastic states in the stationary environment. Since those synapses at less plastic states are hard to modify its strength, the fluctuations in the synaptic strength becomes smaller.

We also found, however, that this desirable property of memory consolidation also leads to a problem of resetting memory. In other words, the cascade model fails to respond to a sudden, step-like change in the environment (*Figure 2B,D*). This is because after staying in a stable environment, many of the synapses are already in deeper, less plastic, states of cascade. In fact, as seen in *Figure 2D*, the time required to adapt to a new environment increases proportionally to the duration of the previous stable environment. In other words, what is missing in the original cascade model is the ability to reset the memory, or to increase the rate of plasticity in response to an unexpected change in the environment. Indeed, recent human experiments suggest that humans can react to such sudden changes by increasing their learning rates (*Nassar et al., 2010*).

To overcome this problem, we introduce a novel surprise detection system with plastic synapses that can accumulate reward information and monitor the performance of decision-making network over multiple (discrete) timescales. The main idea is to compare the reward information of multiple timescales that are stored in plastic (but not meta-plastic) synapses in order to detect changes on a trial-by-trial basis. More precisely, the system compares the current difference in reward rates between a pair of timescales to the expected difference; once the former significantly exceeds the latter, a surprise signal is sent to the decision making network to increase the rate of synaptic plasticity in the cascade models.

The mechanism is illustrated in *Figure 2E–H*. The synapses in this system follow the same reward based learning rules as in the decision making network. The important difference, however, is that unlike the cascade model, the rate of plasticity is fixed, and each group of synapses takes one of the logarithmically segregated rates of plasticity $\alpha_i$'s (*Figure 2E*). Also, the learning takes place *independent of* selected actions in order to monitor the overall performance. While the same computation is performed on various pairs of timescales, for illustrative purposes only the synapses belonging to two timescales are shown in *Figure 2G*, where they learn the reward rates on two different timescales by two different rates of plasticity (say, $\alpha_i$ and $\alpha_j$ and $\alpha_i \gg \alpha_j$ ). As can be seen, when the environment and incoming reward rate is stable, the estimate of the more plastic population fluctuates around the estimate of the less plastic population within a certain range. This fluctuation is *expected* from the past, since the rewards were delivered stochastically, but the probability was well estimated. This expected range of fluctuation is learned by the system by simply integrating the difference between the two estimates with a learning rate $\alpha_j$, which we call *expected uncertainty*, inspired by (*Yu and Dayan, 2005*) (the shaded area in *Figure 2G*). Similarly, we call the current difference in the two estimates *unexpected uncertainty* (*Yu and Dayan, 2005*). Updating unexpected uncertainty involves a prediction error signal, which is the difference between the unexpected uncertainty and the current expected uncertainty.

If the unexpected uncertainty significantly exceeds the expected uncertainty (indicated by yellow in *Figure 2G*), a surprise signal is sent to the decision making network, resulting in an increase in the plasticity of the cascade model synapses; thus, the synapses increase their transition rates between depressed and potentiated states. We allow this to take place in the states higher (or more plastic) than $j$ ($k \leq j$). This selective modification is not crucial in a simple task but may become important in more complex tasks in order to retain information on longer timescales that is still useful, such as task structures or cue identities. As encoding these information is in fact beyond the limit of our simple decision making network, we leave this study for future works. The surprise signal is transmitted as long as the unexpected uncertainty significantly exceeds the expected uncertainty, during which the synapses that received the surprise signal keep enhanced plasticity rates so that they reset the memory (*Figure 2H*). Ultimately, expected uncertainty catches up with unexpected uncertainty so that synapses can start consolidating the memory again with the original cascade model transition rates.

Thanks to the surprise detection system, the decision making network with cascade model synapses can now adapt to an unexpected change. As seen in *Figure 2C,D,F*, it can successfully achieve both consolidation (i.e. accurate estimation of probabilities before the change point) and the quick adaptation to unpredicted changes in the environment. This is because the synapses can gradually consolidate the values by becoming less plastic as long as the environment is stationary, while plasticity can be boosted when there is a surprise signal so that memory can be reset. This can be seen prominently in *Figure 2H*, where the distribution of synaptic plasticity decreases over time before the change point, but increases afterwards due to the surprise signal.

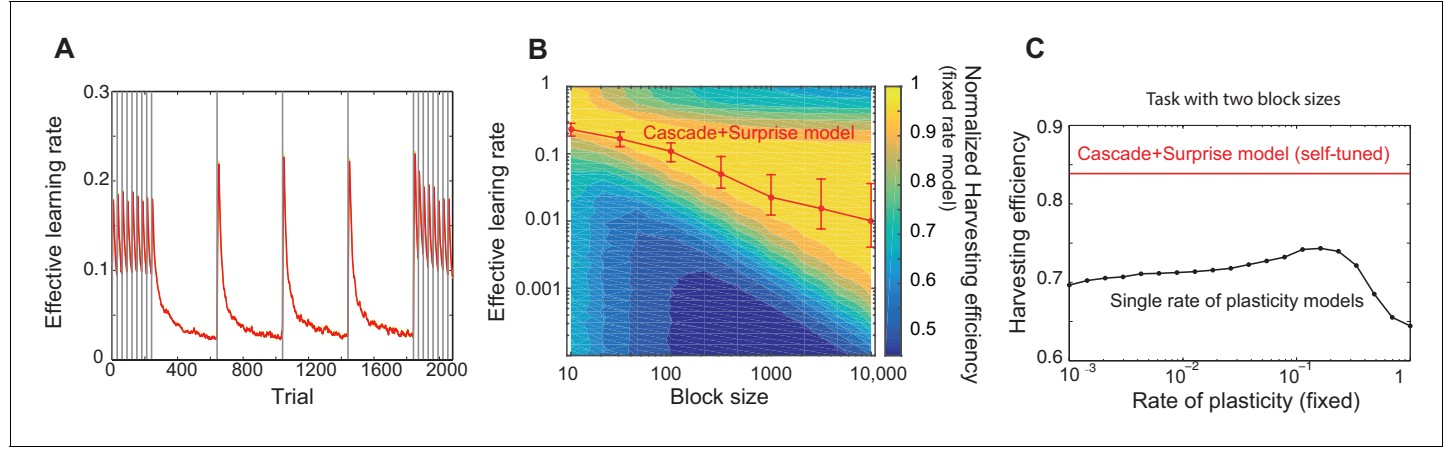

**Figure 3.** Our model captures key experimental findings and it shows a remarkable performance with little parameter tuning. (**A**) The effective learning rate (red), defined by the average potentiation/depression rate weighted by the synaptic population on each state, changes depending on the volatility of the environment, consistent with key experimental findings in *Behrens et al. (2007)*, *Nassar et al. (2010)*. The learning rate gradually decreases over each stable condition, while it rapidly increases in response to a sudden change in environment. The grey vertical lines indicate the change points of contingencies. (**B**) The effective learning rate is self-tuned depending on the timescale of the environment. This contrasts the effective learning rate of our model (red line) to the harvesting efficiency if the model had a single-fixed rate of plasticity in a multi-armed bandit task with given block size (indicated by x-axis). The background colour shows the normalized harvesting efficiency of a single rate of plasticity model, which is defined by the amount of rewards that the model collected, divided by the maximum amount of rewards that the best model for each block size collected, so that the maximum is always equal to one. The median of the effective learning rate in each block is shown by the red trace, as the effective learning rate constantly changes over trials. The error bars indicate the 25th and 70th percentiles of the effective learning rates. (**C**) Our cascade model of metaplastic synapses can significantly outperform the model with fixed learning rates when the environment changes on multiple timescales. The harvest efficiency of our model of cascade synapses combined with surprise detection system (red) is significantly higher then the ones of the model with fixed learning rates, or the rates of plasticity (black). The task is a four-armed bandit task with blocks of 10 trials and 10,000 trials with the total reward rate = 1. The total number of blocks is set to 1000 : 1. In a given block, one of the targets has the reward probability of 0.8, while the others have 0.2. The network parameters are taken as $\alpha_r^i = 0.5^i$, $\alpha_{nr}^i = 0.5^{i+1}$, $p_r^i = 0.5^i$, $p_{nr}^i = 0.5^{i+1}$, $T = 0.1$, $\gamma = 1$, $m = 12$, $h = 0.05$ for (**A**), $\alpha_i = p_i = 0.5^i$, $T = 0.1$, $\gamma = 1$, $m = 20$, $h = 0.05$ for (**B**), $\alpha_i = p_i = 0.5^i$, $T = 0.1$, $\gamma = 1$, $m = 4$, $h = 0.0005$ for (**C**), and $\gamma = 1$ and $T = 0.1$ for the single timescale model in (**B**).

For more details of implementation of our model, including how the two systems work as a whole, please see the Materials and methods section and Figure 8 wherein.

## Our model self-tunes the learning rate and captures key experimental findings

Experimental evidence shows that humans have a remarkable ability to change their learning rates depending on the volatility of their environment (*Behrens et al., 2007*; *Nassar et al., 2010*). Here we show that our model can capture this key experimental finding. We note that single learning rates have been usually reported in most of the past analyses of experimental data. This was simply because single timescale models were assumed when fitting data. Our model, however, has no specific timescale, since it has a wide range of timescales in metaplastic states. Thus, merely for the purpose of comparison of our results with previous findings from single timescale models, we define the *effective* learning rate of our system as the average transition rates $\alpha_i$'s weighted by the synaptic populations that fill corresponding states. Changes in learning rate were therefore characterized by changes in the distribution in synaptic plasticity states in our model.

In *Figure 3A*, we simulated our model in a four-armed bandit task, where one target has a higher probability of obtaining reward than the other targets, while the identity of the most rewarding target is switched at the change points indicated by vertical lines. We found that the effective learning rate is on average significantly larger when the environment is rapidly changing (those trials in shorter blocks) than when the environment is more stable (those trials in longer blocks). This is consistent with the experimental finding in (*Behrens et al., 2007*) that the learning rate was high in a smaller block (volatile) condition than in a larger block (stable) condition. Also, within each block of trials, we found that the learning rate is largest after the change point, decaying slowly over subsequent trials. This is consistent with both experimental findings and the predictions of optimal Bayesian models (*Nassar et al., 2010*; *Dayan et al., 2000*).

It should be noted that our model does not assume any *a priori* timescale of the environment. Rather, the distribution of the rates of synaptic plasticity is dynamically self-tuned to a given environment. To see how well the tuning is achieved, in *Figure 3B*, we contrasted the effective learning rate of our model (red line) under a fixed block size condition (the size was varied over x-axis), to the harvesting efficiency of a single timescale model with different rates of plasticity (varied over y-axis, which we simply call here the learning rate). The background colour shows the normalized harvesting efficiency of single rate of plasticity models, which is defined by the amount of rewards that the model collected, divided by the maximum amount of rewards that the best model for each block size collected, so that the maximum is always equal to one. The effective learning rate of our full model is again defined by the average potentiation/depression rate weighted by the synaptic population on each state, and the median of the effective learning rate in each block is shown by the red trace. (Note that the effective learning rate constantly changes over trials. The error bars indicate the 25th and 70th percentiles of the effective learning rates.) As can be seen, the cascade model's effective leaning rate is automatically tuned to the learning rate expected from the hand-tuned non-cascade plasticity model. This agreement is remarkable, as we did not assume any specific timescales in our cascade model of plasticity nor any optimisation technique; rather, we assumed a wide range of timescales ($1/\alpha_i$'s) and that synapses make reward-based plastic and metaplastic transitions by themselves, guided by surprise signals.

Moreover, we found that our cascade model of metaplastic synapses can significantly outperform the model with fixed learning rates when the environment changes on multiple timescales, which is a very realistic situation but has yet to been explored experimentally. We simulated a four-armed bandit task with two different sizes of blocks with fixed reward contingencies, which is similar to the example in *Figure 3A*. As seen in *Figure 3C*, our model of cascade synapses combined with surprise detection system can collect significantly more rewards than any model with fixed single synaptic plasticity. This is because that the synaptic plasticity distribution of the cascade model is self-tuned on a trial-by-trial basis, rather than on average over a long timescale, as shown in *Figure 3A*. We also found that this is true with a very wide range of threshold values for the surprise detection network, indicating that tuning of the threshold is not required.

In order to further investigate the optimality of our neural model, we compared our model with a previously proposed Bayesian learner model (*Behrens et al., 2007*). This Bayesian model has been proposed to perform an optimal inference of changing reward probabilities and the volatility of the

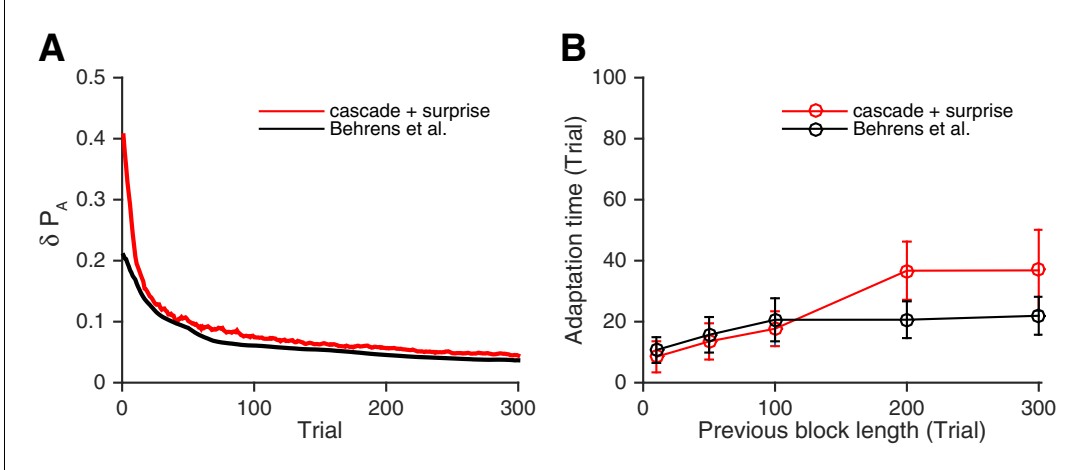

**Figure 4.** Our neural circuit model performs as well as a previously proposed Bayesian inference model (*Behrens et al., 2007*). (**A**) Changes in the fluctuation of choice probability in a stable environment. As shown in previous figures, our cascade model synapses with a surprise detection system (red) reduces the fluctuation gradually over time. This is also the case for the Bayesian model (black). Remarkably, our model reduces the fluctuation as fast as the Bayesian model (*Behrens et al., 2007*). The probability fluctuation $\delta P_A$ is defined as a mean standard deviation in the simulated choice probabilities. The synapses are assumed to be at the most plastic states at $t = 0$, and uniform prior was assumed for the Bayesian model at $t = 0$. (**B**) The adaptation time required to switch to a new environment after a change point. Again, our model (red) performs as well as the Bayes optimal model (black). Here the adaptation time $\tau$ is defined as the number of trials required to cross the threshold probability ($P_A = 0.6$) after the change point. The task is a 2-target VI schedule task with the total baiting rate of $= 0.4$. The network parameters are taken as $\alpha_i = 0.2^i$, $p_i = 0.2^i$, $T = 0.1$, and $\gamma = 0$, $m = 10$, $h = 0.01$. See Materials and methods, for details of the Bayesian model.

environment. While human behavioral data has been shown to be consistent with what the optimal model predicted (*Behrens et al., 2007*), this model itself, however, does not account for how such an adaptive learning can be achieved neurally. Since our model is focused on an implementation of adaptive learning, a comparison of our model and the Bayes optimal model can address this issue.

For this purpose, we simulated the Bayesian model (*Behrens et al., 2007*), and compared the results with our model's results. Remarkably, as seen in *Figure 4*, we found that our neural model (red) performed as well as the Bayesian learner model (black). *Figure 4A* contrasts the fluctuation of choice probability of our model to the Bayesian learner model under a fixed reward contingency. As seen, the reduction of fluctuations over trials in our model is strikingly similar to that the Bayesian model predicts. *Figure 4B*, on the other hand, shows the adaptation time as a function of the previous block size. Again, our model performed as well as the Bayesian model across conditions, though our model was marginally slower than the Bayesian model when the block was longer. (Whether this small difference in the longer block size actually reflects biological adaptation or not should be tested in future experiments, as there have been limited studies with a block size in this range.)

So far we have focused on changes in learning rate; however, our model has a range of potential applications to other experimental data. For example, here we briefly illustrate how our model can account for a well-documented phenomenon that is often referred to as the spontaneous recovery of preference (*Mazur, 1996*; *Gallistel et al., 2001*; *Rescorla, 2004*; *Lloyd and Leslie, 2013*). In one example of animal experiments (*Mazur, 1996*), pigeons performed an alternative choice task on a variable interval schedule. In the first session, two targets had the same probability of rewards. In the following sessions, one of the targets was always associated with a higher reward probability than the other. In these sessions, subjects showed a bias from the first session persistently over multiple sessions, most pertinently in the beginning of each session. Crucially, this bias was modulated by the length of inter-session-intervals (ISIs). When birds had long ISIs, the bias effect was smaller and the adaptation was faster. One idea is that subjects 'forget' recent reward contingencies during long ISIs.

We simulated our model in this experimental setting, and found that our model can account for this phenomenon (*Figure 5*). The task consists of four sessions, the first of which had the same probability of rewards for two targets (3000 trials). In the following sessions, one of the targets (target A)

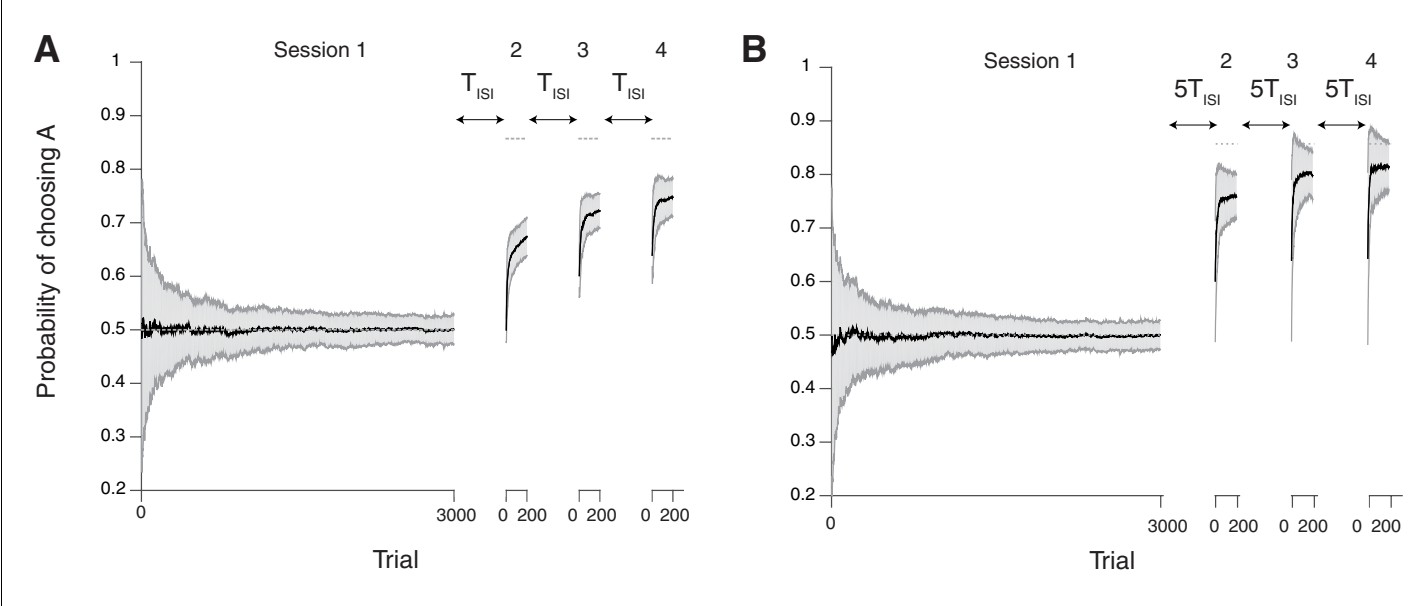

**Figure 5.** Our neural model with cascade synapses captures spontaneous recovery of preference (*Mazur, 1996*). (**A**) Results for short inter-session-intervals (ISIs) (= 1 $T_{\mathrm{ISI}}$). (**B**) Results for long ISIs (= 5 $T_{\mathrm{ISI}}$). In both conditions, subjects first experience a long session (Session 1 with 3000 trials) with a balanced reward contingency, then following sessions (Sessions 2,3,4, each with 200 trials) with a reward contingency that is always biased toward target A (reward probability ratio: 9 to 1). Sessions are separated by ISIs, which we modeled as a period of forgetting according to the rates of plasticity in the cascade model (see *Figure 7*). As reported in (*Mazur, 1996*), the overall adaptation to the new contingency over sessions 2–4 was more gradual for short ISIs than long ISIs. Also, after each ISI the preference dropped back closer to the chance level due to forgetting of short timescales; however, with shorter ISIs subjects were slower to adapt during sessions. The task is a alternative choice task on concurrent VI schedule with the total baiting rate of 0.4. The mean and standard deviation of many simulation results are shown in Black line and gray area, respectively. The dotted horizontal lines indicate the target choice probability predicted by the matching law. The network parameters are taken as $\alpha_i = 0.2^i$, $p_i = 0.2^i$, $T = 0.1$, and $\gamma = 0$, $m = 10$, $h = 0.001$.

was always associated with a higher reward probability than the other (the reward ratio is 9 to 1; 200 trials per session). We simulated our model in a task with short (*Figure 5A*) and long (*Figure 5B*) ISIs. We assumed that the cascade model synapses 'forget' during the ISI, simulated by random transitions with the probabilities according to each synaptic states (See Materials and methods and Figure 7).

As seen in *Figure 5*, the model shows a bias from the first session persistently over multiple sessions (Sessions 2–4), most pertinently in the beginning of each session. Also, learning was slower with shorter ISIs, which is consistent with findings in *Mazur (1996)*. This is because the cascade model makes metaplastic transitions to deeper states (memory consolidation) during stable session 1, and those synapses are less likely to be modified in later sessions, remaining as a bias. However, they could be reset during each ISI due to forgetting transitions (Figure 7), the chance of which is higher with a longer ISI.

We also found that the surprise system played little role in this spontaneous recovery, because forgetting during the ISI allowed many synapses to become plastic, a function virtually similar to what the surprise system does at a block change in block-designed experiments. Crucially, however, not all synapses become plastic during the ISIs, leading to a persistent bias toward the previous preference. Our model in fact predicts such a bias can develop over multiple sessions, and this is supported by experimental data (*Iigaya et al., 2013*). We plan to present this formally elsewhere. Also, we note that our model echoes with the idea that animals carry over memory of contexts of the first session to later sessions (*Lloyd and Leslie, 2013*).

# Discussion

Humans and other animals have a remarkable ability to adapt to a changing environment. The neural circuit mechanism underling such behavioral adaptation has remained, however, largely unknown. While one might imagine that the circuits underlying such remarkable flexibility must be very complex, the current work suggests that a relatively simple, well-studied decision-making network, when combined with a relatively simple model of synaptic plasticity guided by a surprise detection system, can capture a wide range of existing data.

We should stress that there have been extensive studies of modulation of learning in conditioning tasks in psychology, inspired by two very influential proposals. The first was by Mackintosh (*Mackintosh, 1975*), in which he proposed that learning should be enhanced if a stimulus predicts rewards. In other words, a reward-irrelevant stimulus should be ignored, while a reward-predictive stimulus should continue to be attended to. This can be interpreted in our model in terms of formations of stimulus-selective neural populations in the decision making circuit. In other words, such a process would be equated with a shaping of the network architecture itself. This modification is beyond the scope of the current work, and we leave it as future work. The other influential proposal was made by Pearce and Hall (*Pearce and Hall, 1980*). They proposed that learning rates should be increased when an outcome was unexpected. This indeed is at the heart of the model proposed here, where unexpected uncertainty enhanced synaptic plasticity and hence the learning rate. Since the Pearce-Hall model focused on the algorithmic level of computation while our work focusing more on neural implementation level of computation, our work complements the classical model of Pearce and Hall (*Pearce and Hall, 1980*). We should, however, stress again that how our surprise detection system can be implemented should still be determined in the future.

In relation to surprise, the problem of change-point detection has long been studied in relation to the modulation of learning rates in reinforcement learning theory and Bayesian optimal learning theory (*Pearce and Hall, 1980*; *Adams and MacKay, 2007*; *Dayan et al., 2000*; *Gallistel et al., 2001*; *Courville et al., 2006*; *Yu and Dayan, 2005*; *Behrens et al., 2007*; *Summerfield et al., 2011*; *Pearson and Platt, 2013*; *Wilson et al., 2013*). These models, however, provided limited insight into how the algorithms can be implemented in neural circuits. To fill this gap, we proposed a computation which is partially performed by bounded synapses, and we found that our model performs as well as a Bayesian learner model (*Behrens et al., 2007*). We should, however, note that we did not specify a network architecture for our surprise detection system. A detailed architecture for this, including connectivity between neuronal populations, requires more experimental evidence. For example, how the difference in reward rates (subtraction) were computed in the network needs to be further explored theoretically and experimentally. One possibility is a network that includes two neural populations (X and Y), each of whose activity is proportional to its synaptic weights. Then one way to perform subtraction between these populations would be to have a readout population that receives an inhibitory projection from one population (X) and an excitatory projection from the other population (Y). The activity of the readout neurons would then reflect the subtraction of signals that are proportional to synaptic weights (Y–X).

Nonetheless, the surprise detection algorithm that we propose was previously hinted by Aston and Cohen (*Aston-Jones and Cohen, 2005*), where they suggested that task-relevant values computed in the anterior cingulate cortex (ACC) and the orbitofrontal cortex (OFC) are somehow integrated on multiple timescales and combined at the locus coeruleus (LC), as they proposed that the phasic and tonic release of norepinephrine (NE) controls the exploitation-exploration tradeoff. Here we showed that this computation can be carried out mainly by synaptic plasticity. We also related our computation to the notions of unexpected and expected uncertainties, which have been suggested to be correlated with NE and Acetylcholine (Ach) release, respectively (*Yu and Dayan, 2005*). In fact, there is increasing evidence that the activity of ACC relates to the volatility of the environment (*Behrens et al., 2007*) or surprise signal (*Hayden et al., 2011*). Also, there is a large amount of experimental evidence that Ach can enhance synaptic plasticity (*Gordon et al., 2005*; *Mitsushima et al., 2013*). This could imply that our surprise signal could be expressed as the balance between Ach and NE. On the other hand, in relation to encoding reward history over multiple timescales, it is well known that the phasic activity of dopaminergic neurons reflects a reward prediction error (*Schultz et al., 1997*), while tonic dopamine levels may reflect reward rates (*Niv et al., 2007*); these signals could also play crucial roles in our multiple timescales of reward integration

process. We also note that a similar algorithm for the surprise detection was recently suggested in a reduced Bayesian framework (*Wilson et al., 2013*).

In this paper, we assume that the surprise signals are sent when the incoming reward rate decreases unexpectedly, so that the cascade model synapses can increase the rate of plasticity and reset memory. However, there are other cases where surprise signals could be sent to modify the rates of plasticity. For example, when the incoming reward rate is dramatically increased, surprise signals could enhance the metaplastic transitions so that the memory of recent action values are rapidly consolidated. Also, in response to an unexpected punishment rather than reward, surprise signals could be sent to enhance the metaplastic transitions to achieve a one-shot memory (*Schafe et al., 2001*). Furthermore, the effect of the surprise signal may not be limited to reward-based learning. An unexpected recall of episodic memory could itself also trigger a surprise signal. This may explain some aspects of memory reconsolidation (*Schafe et al., 2001*).

Our model has some limitations. First, we mainly focused on a relatively simple decision making task, where one of the targets is more rewarding than the other and the reward rates for targets change at the same time. In reality, however, it is also possible that reward rates of different targets change independently. In this case it would be preferable to selectively change learning rates for different targets, which might be solved by incorporating an additional mechanism such as synaptic tagging (*Clopath et al., 2008*; *Barrett et al., 2009*). Second, although we assumed that the surprise signal would reset most of the accumulated evidence when reward-harvesting performance deteriorates, in many cases it would be better to keep accumulated evidence, such as to form distinct 'contexts' (*Gershman et al., 2010*; *Lloyd and Leslie, 2013*). This would allow subjects to access it later. This type of operation may require further neural populations to be added to the decision making circuit that we studied. In fact, it has been shown that introducing neurons that are randomly connected to neurons in the decision making network can solve context dependent decision-making tasks (*Rigotti et al., 2010*; *Barak et al., 2013*). Those randomly connected neurons were reported in the prefrontal cortex (PFC) as 'mixed-selective' neurons (*Rigotti et al., 2013*). It would be interesting to introduce such neuronal populations to our model to study more complex tasks.

Also, distributing memory among different brain areas may also allow flexible access of memory on different timescales, or hierarchical structure of contexts, if the rates of synaptic plasticity are similarly distributed amongst different brain areas, with memory information being transferred from one area to another (*Squire and Wixted, 2011*). Indeed, it has recently been shown that such a partitioning could also be advantageous general memory performance (*Roxin and Fusi, 2013*), and this could be incorporated with relative ease into our model. One possibility is that the value signals computed by the cascade model synapses with a different range of timescales in distinct brain areas are combined to make decisions, so that the surprise signal is sent to the appropriate brain areas with the targeted rates of plasticity and contexts.

Some of the key features of our model remain to be tested as predictions. One is that the synapses encoding action values in the decision making network should change the level of plasticity itself. In other words, those synapses that reach the boundary of synaptic strength should become more resilient to change. For example, if rewards are given every trial from the same target, the synaptic strength targeting such target would reach the boundary, say after 100 trials. This means that the synaptic strength would remain the same, even after 1000 trials. However, the synapses after 1000 trials should be more resilient to change than synapse after 100 trials. Equally, the synapses that encode overall reward rates, or subject's performance, in a surprise detection system should not make meta-plastic transitions. Thus, studying the nature of synaptic plasticity may allow us to dissociate the functions of circuits.

While we found that our model is robust to parameter changes, the effect of extreme parameter values may give insights into psychiatric and personality disorders. For example, if the threshold of the surprise signal, $h$, is extremely low, the model can become inflexible in the face of changes in the environment. On the other hand, if the threshold is extremely high, the model cannot consolidate the values of actions, leading to unstable behavior. As these sorts of maladaptive behaviors are common across different psychiatric and personality disorders, our model could potentially provide insights into the circuit level dynamics underlying aspects of these disorders (*Deisseroth, 2014*).

# Materials and methods

Our model consists of two systems: (1) the decision making network, which makes decisions according the actions values stored in plastic synapses (2) the surprise detection system, which computes expected uncertainties and unexpected uncertainties on multiple timescales to send a surprise signal to the decision making network, when the unexpected uncertainty exceeds the expected uncertainty.

## The decision making network with cascade type synapses

The decision making network (*Soltani and Wang, 2006*; *Fusi et al., 2007*; *Wang, 2008*; *Iigaya and Fusi, 2013*) is illustrated in *Figure 1A*. In this neural circuit, two groups of excitatory neurons (decision populations), each of which is selective to an action of choosing each target stimuli (A or B), receive inputs from sensory neurons on each trial. Each of the excitatory populations are recurrently connected to sustain their activity during each trial. In addition, they inhibit with each other through a inhibitory neuronal population.

As a result of the inhibitory interaction, the firing rate of one population of excitatory neurons become much larger than the other population (winner take all process) (*Wang, 2002*). This is a stable state of this attractor network, and we assume that subject's action is determined by the winning population (selecting A or B).

Soltani and Wang (*Soltani and Wang, 2006*) showed in simulations of a such network with spiking neurons that the decision of the attractor network is stochastic, but the probability of choosing a particular target can be well fitted by a sigmoid function of the difference between the synaptic input currents $I_A - I_B$ from the sensory neurons to the action selective populations A and B:

$$P_A = 1 - P_B = \frac{1}{e^{-\frac{I_A - I_B}{T}} + 1}, \tag{2}$$

where $P_A$ is the probability of choosing target $A$ and the temperature $T$ is a free parameter determined by the amount of noise in the network.

The afferent currents $I_A$ and $I_B$ are proportional to the synaptic weights between the input population of neurons and the two decision populations of neurons. The current to a neuron that belongs to the decision of selecting target A can be expressed as:

$$I_A = \sum_{j=1}^{N} w_j^A \nu_j \tag{3}$$

where the $\nu_j$'s are the firing rates of the $i$-th neuron (of the total of $N$ neurons) in the input population and $w_j^A$ is the synaptic weight to the population selective to A. An analogous expression holds for the $I_B$ and we assume that $N$ is the same for both populations. Assuming that the firing rates of input population is approximately to be uniform $\nu_j = \nu$, we can simplify the expression of the current:

$$I_A = \sum_{j=1}^{N} w_j^A \nu = \nu N \langle w \rangle_A \tag{4}$$

where $\langle w \rangle_A$ is the average synaptic weight to the population selective to A. Here we can assume $\nu N = 1$ without any loss of generality, as we can rescale $T$ as $T/\nu N \to T$. Also any overlapping of selectivity or any other noise in those two decision making populations can be incorporated to the temperature parameter $T$ in our model.

Following (*Fusi et al., 2005*), the cascade model of synapses assumes that each synaptic strength is binary – either depressed or potentiated, with the value of 0 or 1, respectively. This follows the important constrant of bounded synapses (*Amit and Fusi, 1994*; *Fusi and Abbott, 2007*), and it has been shown that having intermediate strength between 0 and 1 does not significantly improve model's memory performance (*Fusi and Abbott, 2007*) or decision-making behavior (*Iigaya and Fusi, 2013*). In addition, the cascade model of synapses (*Fusi et al., 2005*; *Soltani and Wang, 2006*; *Iigaya and Fusi, 2013*) assumes synapses can take different levels of plasticity. Following (*Iigaya and Fusi, 2013*), we assume there are $m$ states in this dimension.

Instead of simulating the dynamics of all individual synapse, it is more convenient to keep track of the distribution of synapses over the synaptic state space:

$$\sum_{i=1}^{m} F_i^{A-} + \sum_{i=1}^{m} F_i^{A+} = 1, \tag{5}$$

where $F_i^{A-}$ ($F_i^{A+}$) is the fraction of synapses occupying the depressed (potentiated) state at the $i$'th level of the plasticity state in the population targeting the action of choosing $A$. The same can be written for the synapses targeting the neural population selective to target B. As we assume that the synaptic strength is 0 for the depressed states and 1 for the potentiated states, the total (normalized) synaptic strength can be expressed as

$$\langle w \rangle_A = \sum_{i=1}^{m} F_i^{A+}. \tag{6}$$

Again, an analogous relation holds for the synaptic population between the input neurons and the neurons selective to choosing target B.

Hence the action of choosing A or B is determined by the decision making network as:

$$P_A = \frac{1}{e^{-\frac{\sum_{i=1}^{m}\left(F_i^{A+} - F_i^{B+}\right)}{T}} + 1} \tag{7}$$

Thus the decision is biased by the synapses occupying the potentiated states, which reflects the memory of past rewards that is updated according to a learning rule. Here we apply the standard activity dependent reward-based learning rule (*Fusi et al., 2007*; *Soltani and Wang, 2006*;

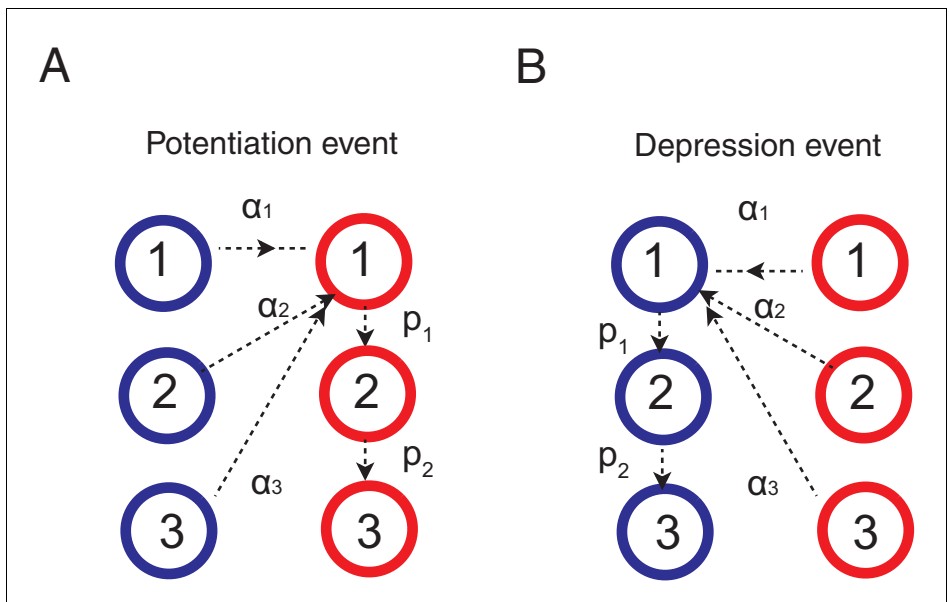

**Figure 6.** Learning rules for the cascade model synapses. (**A**) When a chosen action is rewarded, the cascade model synapses between the input neurons and the neurons targetting the chosen action (hence those that with high firing rates) are potentiated with a probability determined by the current synaptic states. For those synapses at one of the depressed states (blue) would increase the strength and go to the most plastic, potentiated, state (red-1), while those at already one of the potentiated sates (red) would undergo metaplastic transitions (transition to deeper states) and become less plastic, unless they are already at the deepest state (in this example, state 3). (**B**) When an action is not rewarded, the cascade model synapses between the input population and the excitatory population targeting the chosen action are depressed with a probability determined by the current state. One can also assume an opposite learning for the synapses targeting the non-chosen action (In this case, we assume that all transition probabilities are scaled with $\gamma$).

*Soltani et al., 2006*; *Iigaya and Fusi, 2013*) to the cascade model. This is schematically shown in *Figure 6*. When the network received a reward after choosing target A, the synapses between input population and the action selective population that is targeting the just rewarded action A (note that these neurons have a higher firing rates than the other population) make transitions as following.

$$F_1^{A+} \rightarrow F_1^{A+} + \sum_{i=1}^{m} \alpha_r^i F_i^{A-} - p_r^1 F_1^{A+} \tag{8}$$

$$F_{1<i<m}^{A+} \rightarrow F_{1<i<m}^{A+} + p_r^i F_{i-1}^{A+} - p_r^{i+1} F_i^{A+} \tag{9}$$

$$F_m^{A+} \rightarrow F_m^{A+} - p_r^i F_{m-1}^{A+} \tag{10}$$

$$F_{1<i<m}^{A-} \rightarrow F_{1<i<m}^{A-} - \alpha_r^i F_{1<i<m}^{A-} \tag{11}$$

where $\alpha_r^i$ is the transition probability to modify synaptic strength (between depressed 0 and 1) from the $i$'th level to the first level after rewards, and $p_r^i$ is the metaplastic transition probability from $i$'th (upper) level to $i+1$'th (lower) level after a reward. In words, the synapses at depressed states make stochastic transitions to the most plastic potentiated state, while the synapses that were already at potentiated states make stochastic transitions to deeper, or less plastic, states (see *Figure 6*).

For the synapses tarting un-chosen population, we assume the opposite learning:

$$F_1^{B-} \rightarrow F_1^{B-} + \sum_{i=1}^{m} \gamma \alpha_r^i F_i^{B+} - \gamma p_r^1 F_1^{B-} \tag{12}$$

$$F_{1<i<m}^{B-} \rightarrow F_{1<i<m}^{B-} + \gamma p_r^i F_{i-1}^{B-} - \gamma p_r^{i+1} F_i^{B-} \tag{13}$$

$$F_m^{B-} \rightarrow F_m^{B-} + \gamma p_r^i F_{m-1}^{B-} \tag{14}$$

$$F_{1<i<m}^{B+} \rightarrow F_{1<i<m}^{B+} + \gamma \alpha_r^i F_i^{B+} \tag{15}$$

where $\gamma<1$ is the factor determining the probability of chaining states of synapses targeting an unchosen action at a given trial. In words, the synapses at potentiated states make stochastic transitions to the most plastic depressed state, while the synapses that were already at depressed states make stochastic transitions to deeper, or less plastic, states (see *Figure 6*).

Similarly, when the network received no reward after choosing target A, synapses change their states as:

$$F_1^{A-} \rightarrow F_1^{A-} + \sum_{i=1}^{m} \alpha_{nr}^i F_i^{A+} - p_{nr}^1 F_1^{A-} \tag{17}$$

$$F_{1<i<m}^{A-} \rightarrow F_{1<i<m}^{A-} + p_{nr}^i F_{i-1}^{A-} - p_{nr}^{i+1} F_i^{A-} \tag{18}$$

$$F_m^{A-} \rightarrow F_m^{A-} + p_{nr}^i F_{m-1}^{A-} \tag{19}$$

$$F_{1<i<m}^{A-} \rightarrow F_{1<i<m}^{A-} - \alpha_{nr}^i F_{1<i<m}^{A-} \tag{20}$$

and

$$F_1^{B+} \rightarrow F_1^{B+} + \sum_{i=1}^{m} \gamma \alpha_{nr}^i F_i^{B-} - p_{nr}^1 F_1^{B+} \tag{21}$$

$$F_{1<i<m}^{B+} \rightarrow F_{1<i<m}^{B+} + \gamma p_{nr}^i F_{i-1}^{B+} - \gamma p_{nr}^{i+1} F_i^{B+} \tag{22}$$

$$F_m^{B+} \rightarrow F_m^{B+} + \gamma p_{nr}^i F_{m-1}^{B+} \tag{23}$$

$$F_{1<i<m}^{B-} \rightarrow F_{1<i<m}^{B-} - \gamma \alpha_{nr}^i F_{1<i<m}^{B-} \tag{24}$$

where $\alpha_{nr}^i$ is the transition probability from the $i$'th state to the first state in case of no reward, and $p_{nr}^i$ is the metaplastic transition probability from $i$'th (upper) level to $i+1$'th (lower) level after no reward. Unless otherwise noted, in this paper we set $\alpha_n^i = \alpha_{nr}^i (= \alpha_i)$ and $p_n^i = p_{nr}^i (= p_i)$.

In *Figure 5*, we also simulating the effect of inter-session-interval (ISI). To do this, we simply assumed that random noisy events drive forgetting during the ISIs. This was simulated simply by letting synapses undergo what we define as forgetting transitions (*Figure 7*):

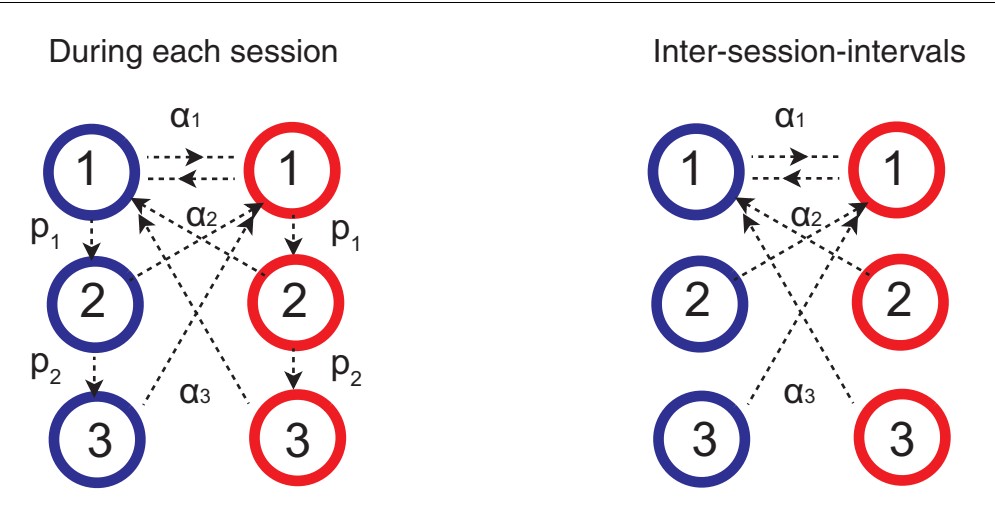

**Figure 7.** Forgetting during inter-session-intervals (ISIs). In our simulations for the spontaneous recovery (*Figure 5*), we assumed that, during the ISI, random forgetting takes place in the cascade model synapses as shown on the right. As a result, synapses at more plastic states were more likely to be reset to the top states. This results in forgetting recent contingency but keeping a bias accumulated over a long timescale.

$$F_1^{A+} \rightarrow F_1^{A+} + \sum_{i=1}^{m} \alpha_i F_i^{A-} \tag{25}$$

$$F_{1<i<m}^{A-} \rightarrow F_{1<i<m}^{A-} - \alpha_i F_{1<i<m}^{A-} \tag{26}$$

and

$$F_1^{A-} \rightarrow F_1^{A-} + \sum_{i=1}^{m} \alpha_i F_i^{A+} \tag{27}$$

$$F_{1<i<m}^{A+} \rightarrow F_{1<i<m}^{A+} - \alpha_i F_{1<i<m}^{A+}. \tag{28}$$

In *Figure 5*, we assume the unit of ISI, $T_{\text{ISI}}$, is 100 repetition of these transitions. We found that our qualitative finding is robust against the setting of threshold value $h$. We did not allow metaplastic (downward) transitions during forgetting, since we focused on the forgetting aspect of ISI, which was sufficient to account for the data (*Mazur, 1996*).

## The surprise detection system

Here we describe our surprise detection system. We do not intend to specify detailed circuit architecture of the surprise detection system. Rather, we propose a simple computation algorithm that can be partially implementable by well-studied bounded synaptic plasticity. As detailed circuits of a surprise detection system have yet to be shown either theoretically or experimentally, we leave a problem of specifying the architecture of system to future studies.

In summary, this system (1) computes reward rates on different timescales (2) computes expected differences between the reward rates of different timescales (we call this as expected uncertainty) (3) compares the expected uncertainty with the current actual difference between reward rates (we call this unexpected uncertainty) (4) sends a surprise signal to the decision making network, if the unexpected uncertainty exceeds the expected uncertainty. As a result, the system receives an input of a reward or no-reward every trial, and sends an output of surprise or no-surprise to the decision making network.

It has been shown that a population of binary synapses can encode the rate of rewards on a timescale of $\tau = 1/\alpha$, where $\alpha$ is the rate of synaptic plasticity (*Rosenthal et al., 2001*; *Iigaya and Fusi, 2013*). Here we use this property to monitor reward rates on multiple timescales, by introducing

populations of synapses with different rates of plasticity. Since the goal of this system is to monitor incoming reward rates on which the cascade model synapses in the decision making network operates, we assume the total of $m$ populations of synapses, where $m$ is the same as the number of meta-plastic states of the cascade model synapses. Accordingly, synapses in population $i$ have the plasticity rate of $\alpha_r^i$, which is the same rate as the cascade model's transition rate at the $i$'th level. Crucially, we assume these synapses are not meta-plastic. They simply undergo reward-dependent stochastic learning; but importantly, this time they do so independent of a chosen action so that the system can keep track of overall performance.

It is again convenient to keep track of the distribution of synapses in the state space. We write the fraction of synapses at the depressed state is $G_i^-$, and the fraction of synapses at potentiated state is $G_i^+$:

$$G_i^- + G_i^+ = 1, \tag{29}$$

Assuming that the synaptic strength is either 0 (depressed) or 1 (potentiated), the total synaptic strength $Z_i$ of population $i$ is simply

$$Z_i = G_i^+ n, \tag{30}$$

where $n$ is the total number of synapses. For simplicity, we assume each population has the same number of synapses. While $Z_i$ is the value that should be read out by a readout, without a loss of generality, we keep track of the normalized weight $R_i = Z_i/n = G_i^+$ as the synaptic strength.

The distribution changes according to a simple reward based plasticity rule (**Iigaya and Fusi, 2013**). When a network receives a reward,

$$G_i^+ \rightarrow G_i^+ + \alpha_r^i G_i^- \tag{31}$$
$$G_i^- \rightarrow G_i^- - \alpha_r^i G_i^-, \tag{32}$$

which means that the synapses at the depressed state make transitions to the potentiated state with a probability of $\alpha_r^i$. When the network received no reward, on the other hand,

$$G_i^- \rightarrow G_i^- + \alpha_{nr}^i G_i^+ \tag{33}$$
$$G_i^+ \rightarrow G_i^+ - \alpha_{nr}^i G_i^+, \tag{34}$$

which means that the synapses at the potentiated state make transitions to the depressed state with a probability of $\alpha_{nr}^i$. The transition rate $\alpha_{nr}^i$ is designed to match the transition rate of the cascade model in case of no-reward. (In this paper we set $\alpha_r^i = \alpha_{nr}^i (= \alpha^i)$, as is also the case in the cascade model synapses in the decision making network.) These transitions take place independent of the taken action, and the synaptic strength $v_i = Z_i/n = G_i^+$ is a low-pass filtered (by bounded synapses) of reward rates on a timescale $\tau_i = 1/\alpha^i$.

On each trial, the system also computes the expected uncertainty $u_{i,j}$ of reward rates between different timescales of synaptic populations. Note that for this we focus on the computational algorithm, and we do not specify the architecture of neural circuits responsible for this computation. As detailed circuits of a surprise detection system have yet to be shown either theoretically or experimentally, we leave a problem of specifying the architecture of system to future studies. The system learns the absolute value of the difference between the approximated reward rates $v_i$ and $v_j$ at a rate of $min(\alpha^j, \alpha^i)$:

$$u_{i,j} = u_{i,j} + min(\alpha^i, \alpha^j)\big(|v_i - v_j| - u_{i,j}\big), \tag{35}$$

where we assume that the learning rate is a smaller rate of plasticity in the two populations. We call $u_{i,j}$ as the expected uncertainty between $i$ and $j$ (**Yu and Dayan, 2005**), representing the how different the reward rates of different timescales are expected to be. We also call the actual current difference $|v_i - v_j|$ as unexpected uncertainty between $i$ and $j$. Hence the expected uncertainty is the low-pass filtered unexpected uncertainty, both of which dynamically change over trials.

On each trial, the system also compares the expected uncertainty $u_{i,j}$ and unexpected uncertainty $|v_i - v_j|$ for each pair of $i$ and $j$. If the latter significantly exceeds the former, $|v_i - v_j| \gg u_{i,j}$, then the

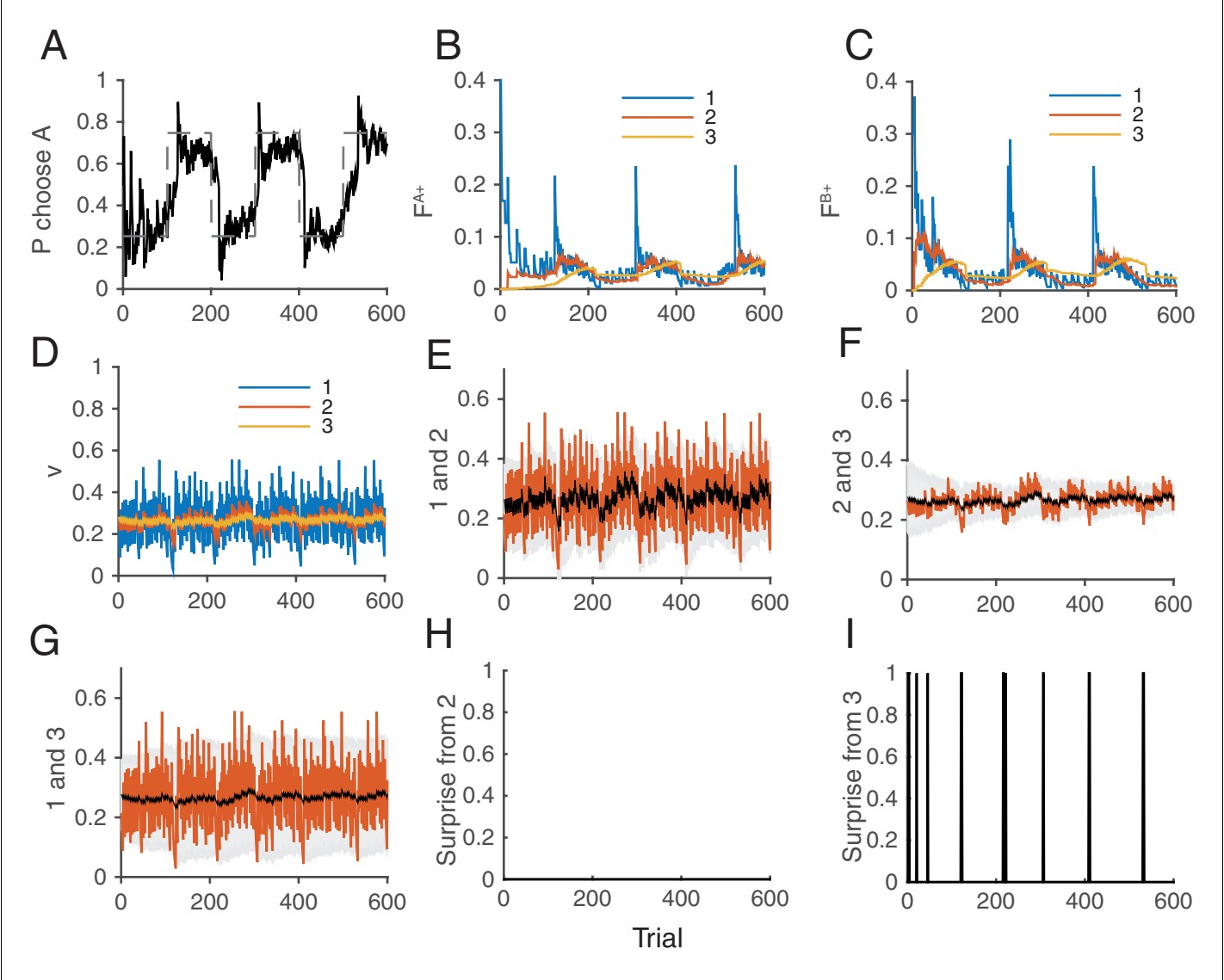

**Figure 8.** How the model works as a whole trial by trial. Our model was simulated on a VI schedule with reward contingency being reversed every 100 trials (between 1 : 4 and 4 : 1). (**A**) The choice probability (solid line) generated from the decision making network. The dashed line indicates the target probability predicted by the matching law. The model's choice probability nicely follows the ideal target probability. (**B**) The distribution of synaptic strength $F_i^{A^+}$ of the population targeting choice A. The different colors indicate different level of the depth $i = 1, 2, 3$ of synaptic states in the cascade model. The sum of these weights give the estimate of the value of choosing A. The shape rises in Blue are due to the surprise signals that were sent roughly every 100 trials due to the block change (see panel I). (**C**) The same for the other synaptic population $F_i^{B^+}$ targeting choice B. (**D**) The normalized synaptic strength $v_i$ in the surprise detection system that integrate reward history on multiple timescales. The numbers for different colors indicate synaptic population $i$, with a fixed rate of plasticity $\alpha_i$. (**E**) The comparison of synaptic strengths $v_i$ between population 1 and 2. The black is the strength of slower synapses $v_2$, while the red is the one of faster synapses $v_1$. The gray area schematically indicates the expected uncertainty. (**F**) The comparison between $v_2$ and $v_3$. (**G**) The comparison between $v_2$ and $v_3$. (**H**) The presence of a surprise signal (indicated by 1 or 0, detected between $v_1$ and $v_2$. There is no surprise since the unexpected uncertainty (red) was within the expected uncertainty (see **E**). (**I**) The presence of a surprise signal detected between $v_1$ and $v_3$, or between $v_2$ and $v_3$. Surprises were detected after each of sudden change in contingency (every 100 trials), mostly between $v_2$ and $v_3$ (see **F**,**G**). This surprise signal enhances the synaptic plasticity in cascade model synapses in the decision making circuit that compute the values of actions shown in **B** and **C**. This enables the rapid adaptation in choice probability seen in **A** The network parameters are taken as $\alpha_i = (\frac{1}{5})^i$, $p_i = (\frac{1}{5})^i$, $T = 0.1$, $\gamma = 0$, $m = 10$, $h = 0.01$.

system sends an output of a surprise signal to the decision making network. For simplicity, we set the threshold $h$ as $erf\left(\frac{v_i-v_j}{\sqrt{2}u_{i,j}}\right)=h$ when $i>j$, where $erf(.)$ is the error function. Note that the error function is sign sensitive. Thus when $v_i>v_j$, or when the reward rate is increasing locally in time, surprise signal is not sent if the threshold is set to be $h<0.5$. This threshold $h$ is a free parameter; but we confirmed that the system is robust over a wide range of $h$.

If a surprise signal is sent, because of the discrepancy between two timescales $i$ and $j$, $|v_i - v_j| \gg u_{i,j}$, the decision making network (cascade synapses) increase the rates of plasticity. Importantly this is done only for the levels of synapses that the surprise is detected (the lower levels do not change the rates of plasticity). This allows the decision-making network to keep information on different timescales as long as it is useful. For example, when a surprise was detected between $i$'th and $j$'th levels, we set the cascade model of transition rates

$$\alpha^k \rightarrow \alpha^1 \qquad (36)$$

for $k \leq j$ of the cascade model synapses. This allows the decision making network to reset the memory and adapt to a new environment. Note that this change of the rate of synapses is only for the cascade model synapses. The synapses in the surprise detection system do not change the rate of plasticity.

*Figure 8* illustrates how the whole system of the decision making network and the surprise detection work together. We simulated our model in a two-choice VI schedule task with a total baiting probability of 0.4. The reward contingency was reversed every 100 trials. The mean synaptic strength of each population $v_i$ is shown in *Figure 8D*, while each pair was compared seperetly in *Figure 8E–G*. Surprises were detected mostly between $v_2$ and $v_3$, or between $v_1$ and $v_3$, (*Figure 8I*), but not between $v_1$ and $v_2$. This makes sense because the timescale of block change was 100 trial, which is similar to the timescale of $v_3$: $1/\alpha_3 = 25$ trials. Thus the timescale of $v_2$ was too short to detect this change: $1/\alpha_3 = 25$ trials. Thanks to the surprise signals, the cascade model of synapses were able to adapt to the sudden changes in contingency (*Figure 8B,C*). As a result, the choice probability also adapt to the environment (*Figure 8A*).

## Bayesian model (*Behrens et al., 2007*)

We also compared our model with a previously proposed Bayesian inference model (*Behrens et al., 2007*). Details of the model can be found in *Behrens et al. (2007)*; thus, here we briefly summarize the formalism. In this model, the probability $R_i^A$ of obtaining a reward from target A at time $t = i$ is assumed to change according to the volatility $v_i^A$.

$$p\left(r_{i+1}^A | r_i^A, v_i^A\right) \sim N\left(r_i^A, V_i^A\right), \qquad (37)$$

where $R_i^A = 1/\left(1 + e^{-r_i^A}\right)$, $V_i^A = e^{v_i^A}$, and $N(,)$ is a Gaussian. Variables are transformed for a computational convenience. The volatility also changes according to the equation:

$$p\left(v_{i+1}^A | v_i^A, k^A\right) \sim N\left(v_{i+1}^A, K^A\right), \qquad (38)$$

where $K^A = e^{k^A}$ determines the rate of change in volatility. Using the Bayes rule, the posterior probability of the joint distribution given data $y^A$ can be written as

$$p\left(r_{i+1}^A, v_{i+1}^A, k | y_{i+1}^A\right) \propto p\left(y_{i+1}^A | r_{i+1}^A\right) \int dr_i^A p\left(r_{i+1}^A | r_i^A, v_i^A\right) \int dv_i^A p\left(r_i^A, v_i^A, k^A | y_i^A\right) p\left(v_{i+1}^A | v_i^A, k^A\right).$$

Following (*Behrens et al., 2007*), we performed a numerical integration over grids without assuming an explicit function form of the joint distribution, where at $t = 0$ we assumed a uniform distribution. Inference was performed for each target independently. For simplicity, we assumed that the model's policy follows the matching law on concurrent VI schedule, as it has been shown to be the optimal probabilistic decision policy (*Sakai and Fukai, 2008*; *Iigaya and Fusi, 2013*).

All the analysis/simulations in this paper were conducted in the MatLab (MathWorks Inc.), and the Mathematica (Wolfram Research).

## Acknowledgements

I especially thank Stefano Fusi for fruitful discussions. I also thank Larry Abbott, Peter Dayan, Kevin Lloyd, Anthony Decostanzo for critical reading of the manuscript; Ken Miller, Yashar Ahmadian, Yonatan Loewenstein, Mattia Rigotti, Wittawat Jitkrittum, Angus Chadwick, and Carlos Stein N Brito for most helpful discussions. I thank the Swartz Foundation and Gatsby Charitable Foundation for generous support.

## Additional information

### Funding

| Funder | Author |
| --- | --- |
| Schwartz foundation | Kiyohito Iigaya |
| Gatsby Charitable Foundation | Kiyohito Iigaya |

The funders had no role in study design, data collection and interpretation, or the decision to submit the work for publication.

### Author contributions

KI, Conception and design, Acquisition of data, Analysis and interpretation of data, Drafting or revising the article

### Author ORCIDs

Kiyohito Iigaya, http://orcid.org/0000-0002-4748-8432

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
