## [Decision Letter]

Thank you for resubmitting your work entitled "Adaptive learning and decision-making under uncertainty by metaplastic synapses guided by a surprise detection system" for further consideration at *eLife*. Your revised article has been favorably evaluated by Eve Marder as the Senior Editor, a Reviewing Editor, and three reviewers.

The author has performed additional simulations and revised the manuscript extensively. All the referees agreed that the manuscript has greatly improved. However, there are some remaining issues to which we would like to see your response.

1) The reviewers pointed out that it is unclear whether the author's model is biologically plausible as proposed. During discussion, however, the reviewers noted that "biophysiological plausibility" is often difficult to define or relative, and that abstract models are often useful. Nevertheless, because the author now emphasizes biological plausibility in order to contrast with existing models (e.g. Bayesian models; Mackintosh; Pearce-Hall), the reviewers thought a little more clarifications or toning down of this point would be required.

We do appreciate that the proposed model is an important step toward a mechanistic investigation of the interesting question; yet, it appears very difficult to implement some of the key components of the model. Specifically, one important proposal is the "surprise detection system" which takes the difference between the current and expected uncertainty, with uncertainty defined as the range of fluctuation (Figure 2). To compute this, the author proposes to calculate the difference in synaptic weights of two groups. This is a very interesting idea yet it is unclear how a neural circuit computes the difference in synaptic weights. One reviewer thought that precisely computing the difference of synaptic weights is beyond the ability of neural circuits (or "out of biological constraints"). We would like you to address this point either by showing how such a computation can be performed or approximated while obeying biological constraints or by simply further de-emphasizing the claim for implementation on specific parts although we note that you already state explicitly that network architecture of the surprise detection system is not specified in the present study, and that the efforts toward biophysical implementation is an important aspect of the present study overall.

2) Please make sure that you do not say that the model "implements" Bayes-optimal solution.

3) One reviewer suggested two additional considerations (Reviewer 1's point #2 and #3). Although we do not see these as essential for revision, they might improve the manuscript. So we would like to see your response.

4) During discussion, all the reviewers agreed that we should not raise the concern of biological plausibility of the cascade model.

Below please find the reviewers' original comments, which contains additional comments for your reference.

*Reviewer #1:*

The author has mostly addressed my comments. Some lingering issues:

1) I don't think it's correct to say that the model implements the Bayes-optimal solution. There's nothing showing that this is true mathematically. What was shown is that it achieves comparable performance. The discussion should be modified to reflect this.

2) The model accounts for the findings of Mazur's second experiment; can it account for the findings of Mazur's first experiment, namely that spontaneous recovery is towards roughly the average of recent sessions? I think it can, which would be a compelling demonstration.

3) While it is nice to see a further application of the model, this seems like a rather random choice of application. Since the author is emphasizing the neural implementation perspective, what one would really like to see is a simulation of specific neural phenomena. Note that the (small number of) phenomena modeled here are all behavioral results. Are there really no neural data bearing on the neural predictions of the model?

*Reviewer #2:*

The manuscript has been significantly improved and also contains new simulation data. I appreciate all these efforts made for improving the clarity of the manuscript. This work shows an interesting idea in computation and will be highly appreciated by computational journals. However, I still doubt whether the model is biologically plausible enough for publication in *eLife*.

The author claims that the model is biologically plausible as it is based on a previously published work of the "cascade synapse model". In fact, I doubt the biological plausibility of the cascade model itself even though the cascade model is unique and provides interesting computational functions. The cascade model assumes binary states to avoid unbounded growth of synaptic strength. However, results from various cortical areas have revealed long-tailed or skewed distributions for the strength of cortical synapses (e.g., Song et al., PLoS Biol 2005; Buzsaki and Mizuseki, Nat Rev Neurosci 2014). These results do not seem to be consistent with binary synapses having only a depressed and a potentiated state. Though the long-tailed distributions contain very strong synapses, these synapses only constitute a small fraction of several thousands of synapses a cortical neuron receives, meaning that the fraction of synapses in the potentiated states should be much smaller than that of synapses in the depressed states. However, it is unclear whether the cascade model, or multi-timescale plasticity, also works under such constraint.

Another concern is that there will be a plenty of different ways to implement a surprise detection system. For example, the detection system may be realized within the framework of reinforcement learning as a system that simply monitors the expected amount of instantaneous reward. Though the author claimed that the previous models of surprised detection did not provide much insight into biological implementation (e.g., in the Discussion), so does the present model. This is my honest impression. I feel that the surprise detection system was proposed in this study just to save the specific cascade model.

*Reviewer #3:*

In the revised paper, several things have been improved.

First, the model by Iigaya is now compared to the Bayesian model by Behrens et al. (2007), and it is shown that the model essentially yields similar results. Second, the model is applied to another type of behavior, and the model can successfully account for this behavior, as well. Third, the method section has been improved and more details to the underpinning of the model have been provided.

In my original review, I had specifically addressed the lack of a clear biophysical implementation of the model. With respect to these points, the author has now more clearly specified the network model, the location of the synapses, and the way they are being modeled. In these respects, I find that the paper has been improved. However, the surprise detection system is still modeled on a purely phenomenological level. This would in principle be fine, except that the author really emphasizes how this model is about a circuit implementation (Marr's third level) of the observed behaviors, and I don’t find that this is really the case.

In fact, my main problem is not even that the surprise detection system is not explicitly modeled as a circuit/ network. Rather, it is that some of the key computations required – taking differences of synaptic strength – seem to rule out *any* halfway realistic circuit computation. How would information about synaptic strength be propagated to reach a location where the subtraction can then be carried out? Apart from wildly speculative ideas, this is not clear to me. The author addressed this by saying that it is left for future work, but the problem is that it looks like this type of computation *cannot* be implemented biophysically. There may be other ways of performing the relevant computations, but the current set of computations really seem to rule out that this could work biophysically.

[Editors’ note: a previous version of this study was rejected after peer review, but the author submitted for reconsideration. The first decision letter after peer review is shown below.]

Thank you for submitting your work entitled "Adaptive learning and decision-making under uncertainty by metaplastic synapses guided by a surprise detection system" for consideration by *eLife*. Your article has been reviewed by three peer reviewers, and the evaluation has been overseen by Naoshige Uchida as Reviewing Editor and Eve Marder as the Senior Editor. Our decision has been reached after consultation between the reviewers. Based on these discussions and the individual reviews below, we regret to inform you that your work will not be considered further for publication in *eLife*.

All the reviewers thought that this work addresses an important question of how the brain adjusts its learning rates in the face of changing volatility of the environment. The author introduce a surprise detection system to a "cascade model" that was previously proposed by Fusi and colleagues. The manuscript is clearly written although it would benefit from better explanations of modeling (see below). Overall, all the reviewers thought that the idea and the results are promising. On the other hand, the reviewers raised a number of concerns that would require substantial revisions. Addressing these concerns would require a substantial amount of simulations and rewriting. It is *eLife*'s policy to not invite revisions that require substantial new scientific work. For that reason, we are forced to reject the manuscript in its current form.

The detailed comments from each referee are attached below. After discussion, the referees thought that the following four points are especially important. First, previous work (e.g. Behrens et al. 2007) have addressed a similar question and presented computational models. The author should compare different models and make the novelty of the current model more explicit. Second, this study only addresses one empirical finding and it is unclear whether this model can explain other phenomena. Applying the current model to other data that demonstrated changes in learning rates would be illuminating. Third, it is argued that the current model is biophysically-inspired but some reviewers thought that the model is still very phenomenological, although this argument could be strengthened by further simulations. Fourth, the methods section requires more work to fully explain the model, and the simulation code should be made available.

*Reviewer #1:*

This paper presents a new computational model of metaplasticity, building on ideas from the cascade model, which allows synapses to rapidly adapt to changing volatility. This is an important question for biological decision-making systems. The article is clearly written and the theory is elegantly simple. However, I have several fundamental concerns that prevent me from recommending this paper for publication.

1) The model only explains a single empirical finding (adaptation of learning rate to reward volatility). This finding is already explained by a number of other models (for example, see Behrens et al. 2007). So it's not clear to me what this new model is adding.

2) While the model is discussed in terms of synapses, no specific biological evidence is presented that directly supports the assumptions of the model.

3) There's a huge literature on the effects of various experimental manipulations on learning rate. Much of this research was inspired by the seminal models of Mackintosh (1975) and Pearce & Hall (1980). Addressing at least some of this literature is important for demonstrating the explanatory power of the model.

*Reviewer #2:*

In this work, Iigaya investigates how organisms can adjust their learning rates to the time scales of a randomly varying and somewhat unpredictable environment. The author studies this problem in the context of models of synaptic plasticity. In these 'cascade' models, learning operates on many different time scales. Iigaya shows that an organism can rapidly switch to the right time scale if it has access to a 'surprise' system that detects any changes in an agents' ability to predict outcomes in the environment. The results are illustrated through various simulations.

Overall, I found the paper quite well written and a pleasure to read. I also think it addresses an interesting and important topic. The only quibble I have is that the model, despite being announced as mechanistic and biophysical, is actually rather phenomenological. It would be nice if the author could find a way to better tie the 'synaptic' plasticity to the underlying neurobiology. For instance, if I were to run an experimental lab and was really interested in these learning questions, what exactly should I measure to test this theory? I elaborate a bit more on this below.

Comments:

1) Biophysical realism: Iigaya emphasizes that this is a model of 'synaptic' plasticity. However, the synapses seem to be considered completely in isolation, and their embedding within a network is only hinted at in words. For instance, no neuron model is specified in the method section, and a (somewhat unspecific) network model is only referenced in the main text. I'd be completely fine with a learning model on a purely phenomenological level. However, if the author wants to emphasize that this type of learning occurs at the level of synapses, he should make the model more biophysical, e.g., by introducing a specific neuron and network model etc. The biophysical plausibility is particularly stretched in equation (24) which learns 'differences' between synaptic weights. I am fine with the learning rules per se, but talking about them in terms of networks and synapses seems a stretch. So either really show that this works within a network, or de-emphasize the biophysical interpretation.

2) One simplification of the whole model seems to be that, if an animal has learnt a particular environment quite well, and synapses are fairly stable with slow plasticity, then a change in environment and the concomitant set of 'surprise signals' would essentially erase everything that had been learnt, and start things from scratch (at least for all learning rates faster than the detected surprise). The author states that longer time scales could remain stable, but it seems to me that does not exactly solve problem of switching between environments each of which changes on a faster time scale. This kind of context-dependence may be worth discussing.

*Reviewer #3:*

Behavioral learning by humans and other animals occurs at multiple timescales. Some years ago, the cascade synapse model successfully modeled the multi-timescale dynamics of synaptic plasticity for decision-making. However, as the overall learning performance gradually shifts to slower timescales in a stationary environment, the cascade synapse model has a difficulty in adapting sudden changes in the environment. To overcome this difficulty, the author proposes a "surprise" detection system for decision-making. The basic idea is to compare the reward information stored in plastic synapses on multiple timescales to detect change points in the environment. Since pieces of evidence suggest that such a signal exists in the brain, the idea and results are of potential interest. However, I feel that the current manuscript is not unambiguously written and is hard to follow for readers unfamiliar with the cascade model. Some improvement is necessary.

Major comments:

1) Figure 2 explains the cascade model and surprise detection system, respectively. While reading the manuscript, I wondered whether the two systems work in harmony or work independently without interactions. Though now I find that the former should be the case, how the two systems interact with one another, or how a surprise signal is informed to the cascade synapses, during decision-making is not perfectly clear to me. Methods also do not clarify my doubt. Please explain more about this point. In Methods, mathematical descriptions also require some revisions. For instance, the definition of *R^B+(-)^* remains unclear in equations. (20-23). Are there also quantities like *R^A+(-)^* as in equations 5-19 of the cascade model? Should the cascade model and surprise detection system have the same depth of multi-timescales? The parameters *α_r_*and *α_nr_* in the r.h.s. of equations 25 and 26 are not defined, and the meaning of these operations is also unclear.

2) Related to the above point, I want to see in Figure 2 how multiple state variables in the cascade model and surprise detector simultaneously evolve on multiple timescales during decision-making. Showing synaptic strength only for two timescales in Figure 2 is not sufficient to understand the entire decision-making system. For example, is a surprise signal detected only at a pair of some timescales or at multiple pairs of different timescales slower than a critical timescale? Does the complex entire system (cascade model + surprise system) always work consistently on all timescales?

3) Results section, subsection “C. Our model self-tunes the learning rate and captures key experimental findings”: The author mentioned that optimal Bayesian model and the proposed model show a similar behavior of the learning rate in each block of trials. Given this information, the readers may wonder what is the advantage of the proposed model over the optimal Bayesian model. Please make comments on this point.

---

## [Author Response]

*The author has performed additional simulations and revised the manuscript extensively. All the referees agreed that the manuscript has greatly improved. However, there are some remaining issues to which we would like to see your response.*

*1) The reviewers pointed out that it is unclear whether the author's model is biologically plausible as proposed. During discussion, however, the reviewers noted that "biophysiological plausibility" is often difficult to define or relative, and that abstract models are often useful. Nevertheless, because the author now emphasizes biological plausibility in order to contrast with existing models (e.g. Bayesian models; Mackintosh; Pearce-Hall), the reviewers thought a little more clarifications or toning down of this point would be required.*

*We do appreciate that the proposed model is an important step toward a mechanistic investigation of the interesting question; yet, it appears very difficult to implement some of the key components of the model. Specifically, one important proposal is the "surprise detection system" which takes the difference between the current and expected uncertainty, with uncertainty defined as the range of fluctuation (Figure 2). To compute this, the author proposes to calculate the difference in synaptic weights of two groups. This is a very interesting idea yet it is unclear how a neural circuit computes the difference in synaptic weights. One reviewer thought that precisely computing the difference of synaptic weights is beyond the ability of neural circuits (or "out of biological constraints"). We would like you to address this point either by showing how such a computation can be performed or approximated while obeying biological constraints or by simply further de-emphasizing the claim for implementation on specific parts although we note that you already state explicitly that network architecture of the surprise detection system is not specified in the present study, and that the efforts toward biophysical implementation is an important aspect of the present study overall.*

Thank you very much for the comments and suggestions. As stated in our original manuscript, we do not intend to propose a network architecture that implements the whole surprise detection algorithm. Specifying the entire architecture will require more experimental evidence and theoretical analysis.

As for the ‘subtraction’, we agree that it is implausible that the system can read out the synaptic strength per se. We sincerely apologize if this caused the confusion. Now we omitted synaptic strength from the following sentence:

In Materials and Methods, in the subsection 'The surprise detection system': “As detailed circuits of a surprise detection system have yet to be shown either theoretically or experimentally, we leave a problem of specifying the architecture of system to future studies. The system learns the absolute value of the difference between the synaptic strength approximated reward rates vi and vj at a rate of…”.

We believe, however, that the difference of weights between two synaptic populations can be approximated by reading out from relevant neural populations. For example, imagine a network that includes two neural populations (A and B), each of whose activity is proportional to its total synaptic weights. Then one way to perform subtraction between these populations would be to have a readout population that receives an inhibitory projection from one population (A) and an excitatory projection from the other population (B). The activity of the readout neurons would then reflect the subtraction of signals that are proportional to synaptic weights (B – A). Now we further emphasize the limitation of the model and mention this possibility:

Second paragraph of Discussion: “We should, however, stress again that how our surprise detection system can be implemented should still be determined in the future.”

“To fill this gap, we proposed a more biophysically implementable computation which is partially performed by bounded synapses, and we found that our model performs as well as a Bayesian learner model”

“We should, however, note that we did not specify a network architecture for our surprise detection system. […] The activity of the readout neurons would then reflect the subtraction of signals that are proportional to synaptic weights (B – A).”

2) Please make sure that you do not say that the model "implements" Bayes-optimal solution.

We sincerely apologize for this. We had no intention to claim that our model implemented the Bayes-optimal solution. We corrected our manuscript to avoid such confusions.

*3) One reviewer suggested two additional considerations (Reviewer 1's point #2 and #3). Although we do not see these as essential for revision, they might improve the manuscript. So we would like to see your response.*

Point #2:

We appreciate this suggestion. We agree that our model would be consistent with the data that the spontaneous recovery was towards the average of recent sessions. In order to further investigate other aspects of spontaneous recovery, including this one, we plan to conduct a more systematic analysis in future studies. Thank you for the suggestion.

Point #3:

Thank you again for the suggestion. Unfortunately, experimental studies into the circuit dynamics of adaptive learning rates are very limited (though, some studies are discussed in the Discussion). As a result, it is currently very difficult to test our model in specific neural data. We hope that our study will stimulate further experimental, and computational, studies.

[Editors’ note: the author responses to the first round of peer review follow.]

*Reviewer #1:*

*This paper presents a new computational model of metaplasticity, building on ideas from the cascade model, which allows synapses to rapidly adapt to changing volatility. This is an important question for biological decision-making systems. The article is clearly written and the theory is elegantly simple. However, I have several fundamental concerns that prevent me from recommending this paper for publication.*

*1) The model only explains a single empirical finding (adaptation of learning rate to reward volatility). This finding is already explained by a number of other models (for example, see Behrens et al. 2007). So it's not clear to me what this new model is adding.*

I’m sorry that it was not clear. We aware that there are models that shows changes in learning rates, including the one by Behrens et al. (2007). However, as we noted above, most computational studies have been limited to Bayesian inference models, which focus on optimal probability interference according to the Bayes law. Those models cannot, by design, specify any biological implementation of such computation. Thus we aimed to provide a more biologically implementable computation in this manuscript, by combining a previously proposed neural circuit model and the cascade model of synaptic plasticity.

We agree that we should compare our model with such optimal computation models. In the current version we simulated Behrens et al. model and compared with our model. We found that our neural model performs as well as the Bayes optimal model. Our results thus now provide a unique insight into how the optimal adaptation of learning rates can be implemented in neural circuits with plastic synapses.

Also, in the current version we account for a different phenomenon with the same model, which is spontaneous recovery of preference.

*2) While the model is discussed in terms of synapses, no specific biological evidence is presented that directly supports the assumptions of the model.*

We apologize that we failed to provide biological supports of the synaptic model. Experiments and computational studies have shown that long time modification of synaptic strengths accounts for memory. It has been recognized, however, that remarkable memory performance of classical memory circuit was based on an assumption of unbounded synaptic weights. Bounding synaptic weights has been shown to create a catastrophic consequence to the memory performance, because synapses ‘forget’ very quickly by overwriting [Amit and Fusi, 1995; Fusi and Abbott, 2007]. However, human memory does not seem to suffer from such a catastrophic forgetting. To account for this, the model of cascade synapse [Fusi et al., 2005] has been proposed. This model was based on the biochemical cascades that are ubiquitous in biological systems and, in particular, are associated with synaptic plasticity. Those processes take place over a wide range of timescales. They showed that the model could significantly improve the model’s memory maintenance performance.

Adaptive decision-making has been studied in a neural circuit model with binary synapses (Soltani and Wang, 2006; Iigaya and Fusi, 2013). The decision-making network was originally proposed by X-J Wang (2002). It is a biophysically based model because it has an “anatomically plausible architecture in which not only single spiking neurons are described biophysically with a reasonable level of accuracy but also synaptic interactions are calibrated by quantitative neurophysiology (which turned out to be critically important) [Wang, 2008]”. It has been shown that the circuit model can account for features of experimental data.

However, it has been recognized that the model has a severe limitation due to the simple synaptic model, that is a speed accuracy trade-off of adaptation. To address this issue, we applied the cascade model of synapses to a well-studied decision-making network.

We did not intend to specify detailed circuit architecture of the surprise detection system. Rather, we proposed a simple computation algorithm that can be partially implementable by a simple binary synaptic plasticity.

As detailed circuits of a surprise detection system have yet to be shown either theoretically or experimentally, we leave a problem of specifying the architecture of system to future studies.

*3) There's a huge literature on the effects of various experimental manipulations on learning rate. Much of this research was inspired by the seminal models of Mackintosh (1975) and Pearce & Hall (1980). Addressing at least some of this literature is important for demonstrating the explanatory power of the model.*

We apologize that we failed to stress the important past research. We discussed these works and their relationship to our work in our current manuscript:

“We should stress that there have been extensive studies of modulation of learning in conditioning tasks in psychology, inspired by two very influential proposals.[…] Since the Pearce-Hall model focused on the algorithmic level of computation while our work focusing on neural implementation level of computation, our work complements the classical model of Pearce and Hall.”

We now also applied our model to a phenomenon called spontaneous recovery, and showed that our model can account for the phenomenon. We should however stress that our work is considered to be complementally to both mackintosh and Pearce & Hall models, because those models do not specify neural implementation of the algorithm. It is David Marr’s 2nd level, algorithm of computation (Marr, 1982). Our approach is at the third level, the neural implementation of computation.

As Marr stressed, these levels should be studied in parallel.

*Reviewer #2:*

*In this work, Iigaya investigates how organisms can adjust their learning rates to the time scales of a randomly varying and somewhat unpredictable environment. The author studies this problem in the context of models of synaptic plasticity. In these 'cascade' models, learning operates on many different time scales. Iigaya shows that an organism can rapidly switch to the right time scale if it has access to a 'surprise' system that detects any changes in an agents' ability to predict outcomes in the environment. The results are illustrated through various simulations.*

*Overall, I found the paper quite well written and a pleasure to read. I also think it addresses an interesting and important topic. The only quibble I have is that the model, despite being announced as mechanistic and biophysical, is actually rather phenomenological. It would be nice if the author could find a way to better tie the 'synaptic' plasticity to the underlying neurobiology. For instance, if I were to run an experimental lab and was really interested in these learning questions, what exactly should I measure to test this theory? I elaborate a bit more on this below.*

*Comments:*

*1) Biophysical realism: Iigaya emphasizes that this is a model of 'synaptic' plasticity. However, the synapses seem to be considered completely in isolation, and their embedding within a network is only hinted at in words. For instance, no neuron model is specified in the method section, and a (somewhat unspecific) network model is only referenced in the main text. I'd be completely fine with a learning model on a purely phenomenological level. However, if the author wants to emphasize that this type of learning occurs at the level of synapses, he should make the model more biophysical, e.g., by introducing a specific neuron and network model etc. The biophysical plausibility is particularly stretched in equation (24) which learns 'differences' between synaptic weights. I am fine with the learning rules per se, but talking about them in terms of networks and synapses seems a stretch. So either really show that this works within a network, or de-emphasize the biophysical interpretation.*

We apologize for this confusion and thank you very much for pointing this out. We now detail this in our new version of manuscript.

The cascade models of synapses are embedded in the X.-J. Wang’s decision-making network. In this network, it has been shown previously that the firing rates of neurons that are responsible for making decisions are largely determined by the strengths of synaptic weights. Hence most of our focus was on the strengths of such synapses. This is now explained in more details in the methods section.

As the reviewer pointed out, however, we did not intend to specify the actual architecture of the other system: the surprise detection system.

This is because there is little experimental and theoretical evidence for specifying the architecture. Hence, for the surprise detection system, we proposed a computational algorithm, which can partially be operated on bounded synapses, without specifying the circuits. We agree that the part that the model learns the difference in the synaptic weights is abstract and we had no intention to specify its biological implementation. We apologize that we did not make this clear. We leave a problem of specifying the architecture of system to future studies. We stress this in the current manuscript:

“Note that for this we focus on the computational algorithm, and we do not specify the architecture of neural circuits responsible for this computation. As detailed circuits of a surprise detection system have yet to be shown either theoretically or experimentally, we leave the problem of specifying the architecture of system to future studies.”

“To fill this gap, we proposed a more biophysically implementable computation, partially performed by bounded synapses, and we found that our model performs as well as a Bayesian learner model (Behrens et al., 2007). We should, however, note that we did not specify network architecture for our surprise detection system. A detailed architecture for this, including connectivity between neuronal populations, requires more experimental evidence.”

*2) One simplification of the whole model seems to be that, if an animal has learnt a particular environment quite well, and synapses are fairly stable with slow plasticity, then a change in environment and the concomitant set of 'surprise signals' would essentially erase everything that had been learnt, and start things from scratch (at least for all learning rates faster than the detected surprise). The author states that longer time scales could remain stable, but it seems to me that does not exactly solve problem of switching between environments each of which changes on a faster time scale. This kind of context-dependence may be worth discussing.*

Thank you very much for pointing this out. This is indeed a limitation of our model. Our model needs a modification to apply more complex situations (for example, what (Gershman et al., 2010) has addressed). In our current manuscript, we explicitly discussed it:

“Our model has some limitations. First, we mainly focused on a relatively simple decision-making task, where one of the targets is more rewarding than the other and the reward rates for targets change at the same time. […] Those randomly connected neurons were reported in PFC as ‘mixed selective’ neurons [50]. It would be interesting to introduce such neuronal populations to our model to study more complex tasks.”

*Reviewer #3:*

*Behavioral learning by humans and other animals occurs at multiple timescales. Some years ago, the cascade synapse model successfully modeled the multi-timescale dynamics of synaptic plasticity for decision-making. However, as the overall learning performance gradually shifts to slower timescales in a stationary environment, the cascade synapse model has a difficulty in adapting sudden changes in the environment. To overcome this difficulty, the author proposes a "surprise" detection system for decision-making. The basic idea is to compare the reward information stored in plastic synapses on multiple timescales to detect change points in the environment. Since pieces of evidence suggest that such a signal exists in the brain, the idea and results are of potential interest. However, I feel that the current manuscript is not unambiguously written and is hard to follow for readers unfamiliar with the cascade model. Some improvement is necessary.*

*Major comments:*

*1) Figure 2 explains the cascade model and surprise detection system, respectively. While reading the manuscript, I wondered whether the two systems work in harmony or work independently without interactions. Though now I find that the former should be the case, how the two systems interact with one another, or how a surprise signal is informed to the cascade synapses, during decision-making is not perfectly clear to me. Methods also do not clarify my doubt. Please explain more about this point. In Methods, mathematical descriptions also require some revisions. For instance, the definition of R^{B+-} remains unclear in equations. (20-23). Are there also quantities like R^{A+-} as in equations 5-19 of the cascade model? Should the cascade model and surprise detection system have the same depth of multi-timescales? The parameters α_{r} and α_{nr} in the r.h.s. of equations 25 and 26 are not defined, and the meaning of these operations is also unclear.*

Thank you very much for pointing this out. We apologize and we detailed this in the Methods section.

*2) Related to the above point, I want to see in Figure 2 how multiple state variables in the cascade model and surprise detector simultaneously evolve on multiple timescales during decision-making. Showing synaptic strength only for two timescales in Figure 2 is not sufficient to understand the entire decision-making system. For example, is a surprise signal detected only at a pair of some timescales or at multiple pairs of different timescales slower than a critical timescale? Does the complex entire system (cascade model + surprise system) always work consistently on all timescales?*

We really appreciate this comment. The whole system is designed to work on all timescales. The cascade model, however, could potentially have a bias to the task relevant time scales. This sometimes leads to a maladaptive behavior when the environment has suddenly changed. To adjust this, the surprise system must operate on all timescales.

It is very important to illustrate how our model works as a whole. We now illustrate this in detail with new Figure 8, and we extended the Methods section.

*3) Results section, subsection “C. Our model self-tunes the learning rate and captures key experimental findings”: The author mentioned that optimal Bayesian model and the proposed model show a similar behavior of the learning rate in each block of trials. Given this information, the readers may wonder what is the advantage of the proposed model over the optimal Bayesian model. Please make comments on this point.*

Thank you very much for pointing this out. As we explained above, we now stress the difference, and conducted an explicit model comparison.